

# Evolution of the patellar sesamoid bone in mammals

Mark E. Samuels[1,2], Sophie Regnault[3] and John R. Hutchinson[3]

[1] Department of Medicine, University of Montreal, Montreal, QC, Canada
[2] Centre de Recherche du CHU Ste-Justine, Montreal, QC, Canada
[3] Department of Comparative Biomedical Sciences, Structure and Motion Laboratory, The Royal Veterinary College, London Hertfordshire, UK

Corresponding author
John R. Hutchinson,
jrhutch@rvc.ac.uk

## ABSTRACT

The patella is a sesamoid bone located in the major extensor tendon of the knee joint, in the hindlimb of many tetrapods. Although numerous aspects of knee morphology are ancient and conserved among most tetrapods, the evolutionary occurrence of an ossified patella is highly variable. Among extant (crown clade) groups it is found in most birds, most lizards, the monotreme mammals and almost all placental mammals, but it is absent in most marsupial mammals as well as many reptiles. Here, we integrate data from the literature and first-hand studies of fossil and recent skeletal remains to reconstruct the evolution of the mammalian patella. We infer that bony patellae most likely evolved between four and six times in crown group Mammalia: in monotremes, in the extinct multituberculates, in one or more stem-mammal genera outside of therian or eutherian mammals and up to three times in therian mammals. Furthermore, an ossified patella was lost several times in mammals, not including those with absent hindlimbs: once or more in marsupials (with some re-acquisition) and at least once in bats. Our inferences about patellar evolution in mammals are reciprocally informed by the existence of several human genetic conditions in which the patella is either absent or severely reduced. Clearly, development of the patella is under close genomic control, although its responsiveness to its mechanical environment is also important (and perhaps variable among taxa). Where a bony patella is present it plays an important role in hindlimb function, especially in resisting gravity by providing an enhanced lever system for the knee joint. Yet the evolutionary origins, persistence and modifications of a patella in diverse groups with widely varying habits and habitats—from digging to running to aquatic, small or large body sizes, bipeds or quadrupeds—remain complex and perplexing, impeding a conclusive synthesis of form, function, development and genetics across mammalian evolution. This meta-analysis takes an initial step toward such a synthesis by collating available data and elucidating areas of promising future inquiry.

## INTRODUCTION

This meta-analysis addresses the evolution of the ossified patella (tibial sesamoid or "kneecap" bone) in mammals. Our focus was on the evolutionary pattern of how bony

patellae evolved in the mammalian lineage, as evidence of osseous patellae is simplest to interpret. However, as explained further below we also consider non-bony sesamoids to also be potential character states of the patellar organ; vexing as the form, fossil record and ontogeny (and thus homology) of those soft-tissue structures are. We compiled voluminous literature and first-hand observational data on the presence or absence of the osseous patella in extinct and extant mammals, then conducted phylogenetic analysis of patellar evolution by mapping these data onto a composite phylogeny of mammals using multiple phylogenetic optimization methods. We used the results to address patterns of acquisition and disappearance (i.e. gain and loss of ossification) of this structure within Mammaliaformes. In particular, we investigated whether an ossified patella was ancestrally present in all crown group Mammalia, and lost in particular groups especially marsupials (Metatheria), or whether it evolved multiple times in separate crown clades. Furthermore, if the bony patella had multiple origins, how many times was it gained or lost and what did it become if it was lost (such as a vestigial fibrocartilage versus complete loss, without any evidence of a sesamoid-like tissue within the patellar tendon)? These were our study's key questions. We provide broader context here first.

Some aspects of the morphology of the knee in tetrapods (four-legged vertebrates bearing limbs with digits) are evolutionarily ancient. Tetrapods had their ancestry amongst lobe-finned sarcopterygian fish, in which jointed, muscular fins transitioned into limbs. Early stages of distinct bony articulations between the femur and tibia–fibula are evident in the hind fins/limbs of Devonian (~370 million years ago; Mya) animals such as *Eusthenopteron*, *Panderichthys* and *Ichthyostega* (*Ahlberg, Clack & Blom, 2005*; *Andrews & Westoll, 1970*; *Boisvert, 2005*; *Dye, 1987*, *2003*; *Haines, 1942*). These fossil sarcopterygians also have subtle differences between the homologous joints in the pectoral fin/forelimb and the pelvic fin/hindlimb, indicating that specification of forelimb/hindlimb identity was already in place (*Boisvert, 2005*; *Daeschler, Shubin & Jenkins, 2006*; *Shubin, Daeschler & Jenkins, 2006*). Furthermore, the morphology of the forelimb and hindlimb joints indicates divergent functions of these limbs, with the forelimb evolving into a more "terrestrialized" capacity earlier than the hindlimb (*Pierce, Clack & Hutchinson, 2012*). Developmental and morphological modifications to the hindlimb and particularly the mid-limb joint between the stylopod and zeugopod continued, until a recognizable knee articulation of almost modern, derived aspect arose in tetrapods of the Carboniferous period, ~350 Mya (*Dye, 2003*).

Sesamoids are best defined as "skeletal elements that develop within a continuous band of regular dense connective tissue (tendon or ligament) adjacent to an articulation or joint" (*Vickaryous & Olson, 2007*). The tibial patella is a sesamoid bone that arises during development within the main extensor tendon of the knee, subsequently "dividing" it (though there remains some continuity) into the quadriceps and patellar tendons (the latter is sometimes inappropriately called the patellar ligament) (*Bland & Ashhurst, 1997*; *Fox, Wanivenhaus & Rodeo, 2012*; *Pearson & Davin, 1921a*; *Tecklenburg et al., 2006*; *Tria & Alicea, 1995*; *Vickaryous & Olson, 2007*). These tendons span from the quadriceps muscle group to the tibia (Fig. 1). The patella itself tends to be incorporated mainly into the

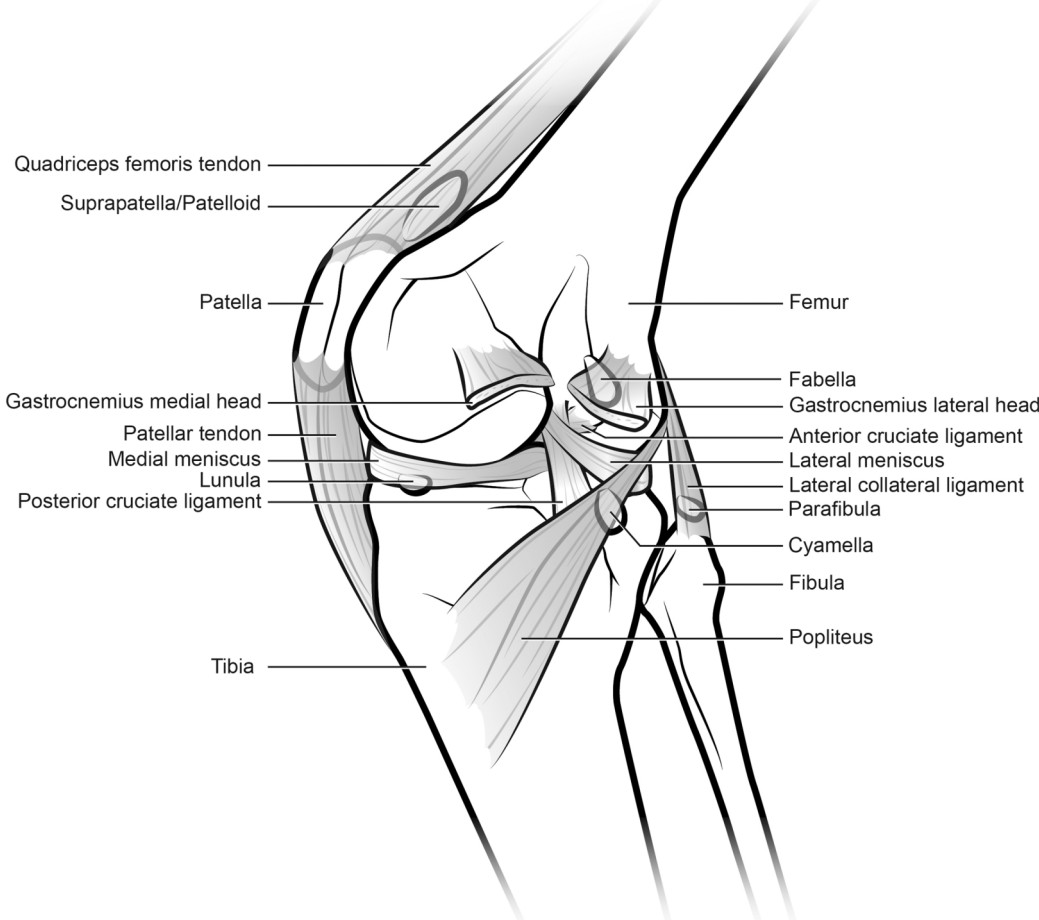

**Figure 1 Generalized knee showing sesamoid bones found in various mammals, although possibly no species includes all of these (patella, lunula, cyamella, fabella and parafibula).** Also shown are relevant muscles, ligaments and other anatomical elements that lie close to the sesamoids of the knee joint. The knee is in medial view and the medial collateral ligament has been removed. Illustration: Manuela Bertoni.

vastus muscles of the quadriceps in mammals, with the tendon of M. rectus femoris lying more superficial to them (*Tria & Alicea, 1995*), with variable degrees of attachment to it (*Jungers, Jouffroy & Stern, 1980*). Hereafter, the term "patella" implies ossification and hindlimb localization unless otherwise specified (some literature inconsistently and confusingly refers to non-ossified cartilaginous structures in this location as patellae—this homology in many cases needs better testing), and implicitly refers to either a single patella or the left and right patellae normally present in an individual. There is an "ulnar patella" in the forelimbs of some taxa (notably lizards, but also some frogs, birds and mammals *Barnett & Lewis, 1958*; *Haines, 1940*; *Maisano, 2002a*, *2002b*; *Pearson & Davin, 1921a*, *1921b*; *Romer, 1976*; *Vanden Berge & Storer, 1995*; *Vickaryous & Olson, 2007*) but a full discussion of this enigmatic structure is beyond the scope of this study. Figure 2 depicts the anatomical orientations used throughout this study to refer to tetrapod limbs.

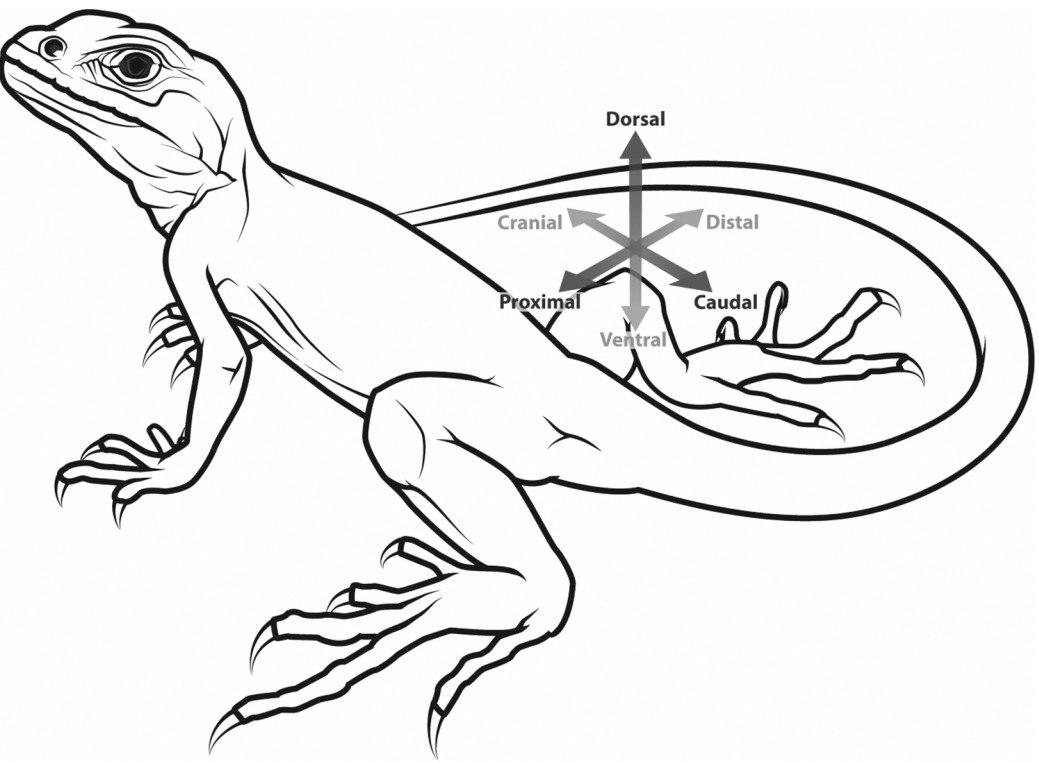

**Figure 2** Generalized tetrapod with anatomical/developmental axes defined for the hindlimb: cranial/caudal (towards the head/tail, respectively), proximal/distal (toward/further from the trunk, respectively), dorsal/ventral (towards the back/belly, respectively). Illustration: Manuela Bertoni.

The patella appears broadly similar amongst mammals possessing it, as far as has been studied, although it varies greatly in size, generally in accordance with body size. It ossifies endochondrally; from a cartilaginous precursor (i.e. anlage *Vickaryous & Olson, 2007*); relatively late in gestation (e.g. sheep, goats *Harris, 1937*; *Parmar et al., 2009*) or sometime after birth (e.g. rabbits, rats, mice, humans *Bland & Ashhurst, 1997*; *Clark & Stechschulte, 1998*; *Patton & Kaufman, 1995*; *Spark & Dawson, 1928*; *Tria & Alicea, 1995*; *Walmsley, 1940*). Very recently, the development of the patella in mouse embryos was re-examined and the claim made that the patella develops as a process that branches off the femur, strongly influenced by mechanical loading in that region (*Eyal et al., 2015*). Whether this truly happens as described in mice, let alone other mammals, and whether it can be accepted as unexpected support for the "traction epiphysis" origin of patellar sesamoids (e.g. *Pearson & Davin, 1921a*, *1921b*), remains to be determined, but the surprising results deserve attention. The general form of the osseous patella in mammals is a hemispherical structure, with a superficial surface (covered by fibrocartilage (*Clark & Stechschulte, 1998*) and quadriceps tendon fibres (*Bland & Ashhurst, 1997*)) and a deep surface which articulates with the femur, gliding along the patellar sulcus or groove in that bone. In maturity, the patella is composed of an outer lamellar cortex enclosing an inner cancellous bone structure with marrow spaces, and has an articular hyaline cartilage lining on the deep surface for articulation with the patellar sulcus (groove) of the femur (*Benjamin et al., 2006*; *Clark & Stechschulte, 1998*; *Vickaryous & Olson, 2007*).

The vastus muscles' tendons (especially M. vastus intermedialis) may have a fibrocartilaginous region at the approximate position of the patella, called the "suprapatella" or "patelloid" (Fig. 1). The latter two terms are sometimes used synonymously, though "suprapatella" is more usual when an osseous patella is also present, and "patelloid" when it is not. The suprapatella is described as proximal to the patella, occasionally with a fat pad interposed between it and the ossified patella (Fig. 1), whilst the patelloid is described as occupying the same approximate region that a bony patella would (though absence of a patella makes this difficult to objectively assess) (*Bland & Ashhurst, 1997*; *Jungers, Jouffroy & Stern, 1980*; *Ralphs, Benjamin & Thornett, 1991*; *Ralphs et al., 1998*; *Ralphs, Tyers & Benjamin, 1992*; *Reese et al., 2001*; *Walji & Fasana, 1983*). It is not clear whether the fibrous patelloid in some marsupials (and perhaps some bats *Smith, Holladay & Smith, 1995*) is homologous to the suprapatella, equivalent to an evolutionarily reduced patella or an independently occurring structure. We revisit this problem later in this study.

The human patellar anlage is first visible at O'Rahilly stage 19, and chondrifies at stage 22. Ossification begins 14 weeks after birth (*Merida-Velasco et al., 1997a*, *1997b*; *Tria & Alicea, 1995*), but is not grossly visible until 4–6 years of age (when multiple, eventually coalescing centres of ossification can be seen radiographically *Ogden, 1984*) and sometimes not in its fully ossified form until adolescence. The patella is the only sesamoid bone counted regularly among the major bones of the human body (*Vickaryous & Olson, 2007*), although there are other, much smaller sesamoids in the hands and feet (and in some cases even the spine; *Scapinelli, 1963*). The pisiform is often considered a sesamoid and deserves further attention in a broad context similar to this study's. Other small sesamoids, such as the lunula, fabella, cyamella and parafibula, also occur in the knee joint in many tetrapod species including some mammals (Fig. 1); these occur sporadically in humans (*Pearson & Davin, 1921a*; *Sarin et al., 1999*).

The patella is covered by the thickest layer of articular cartilage in the human body (*Palastanga, Field & Soames, 2006*). The patella may thus also play a protective role for the underlying joint architecture (*Haines, 1974*), in addition to protecting the patellar tendon from excessive compressive stresses (*Giori, Beaupre & Carter, 1993*; *Sarin & Carter, 2000*; *Wren, Beaupre & Carter, 2000*). The patellar tendon itself, to the extent that its properties are known for some species (e.g. humans), is stiff and strong, able to withstand about twice as much stress as typical knee joint ligaments and enduring strains (i.e. lengthening) of up to 11–14% (*Butler, Kay & Stouffer, 1986*). Regional variations in the microscopic anatomy of the human patella have also been recognized, for example in tissue thickness and nerve arrangement, which may reflect load distribution (*Barton et al., 2007*; *Eckstein, Muller-Gerbl & Putz, 1992*; *Toumi et al., 2006*, *2012*). There is convincing evidence from numerous species that excessive loads on the patella can lead to degeneration of the articular cartilages and damage to the underlying bone, producing osteoarthritis (*Aglietti & Menchetti, 1995*; *Hargrave-Thomas et al., 2013*; *Tria & Alicea, 1995*), so those regional variations of patellar structure are likely important. Similarly, the tissues involved in anchoring the patellar tendon to the proximal and distal surfaces of the patella as well as to the proximal tibia (tuberosity/tubercle) vary in their composition and presumably are

adapted, and exhibit phenotypic plasticity, to reduce the risk of tendon avulsion from the bone (*Evans, Benjamin & Pemberton, 1991*). Reduction of a bony patella to soft tissue presumably reduces its ability to act as a gear or lever (*Alexander & Dimery, 1985*).

Functions of the patella notwithstanding, there was once some enthusiasm for its outright removal for treatment of certain joint problems. Patellectomy was first performed in 1860 and for some time was an established treatment option for several conditions (*Pailthorpe, Milner & Sims, 1991*; *Sweetnam, 1964*). However, partial and complete patellectomies are now considered as last resort salvage procedures; this is also the mainstream view of the veterinary profession (*Langley-Hobbs, 2009*). The historical lack of clarity on the pros and cons of patellectomy was summarized eloquently by *The Lancet*, stating, "Sadly, most of our interventions on the patella are empirical, and are supported more by the enthusiasm of proponents than by a very deep knowledge of the biology or biomechanics of this unusual joint. The knee cap could do with more scientific attention" (*Editors, 1992*).

The latter complaint regarding the dearth of scientific attention to form, development, function and clinical treatment of the patella applies even more so to non-human tetrapods. One exception is a study that measured the inter- and intra-specific variability of the patellae and other bones (*Raymond & Prothero, 2012*). The latter study found generally greater variation in patellae (and other sesamoids) versus "normal" long bones. The inference was that this greater variability might pertain to the "intermembranous" [*sic*-intramembranous] development of sesamoids versus an endochondral location in long bones. However, the patella and most other major limb sesamoids of mammals are pre-formed in cartilage and thus clearly are endochondral bones (*Farnum, 2007*). Yet the latter study (*Raymond & Prothero, 2012*) reinforces that sesamoids are more variable than most other bones, in part due to their mechanical environment, in part due to their embedding in soft tissues (themselves quite variable) such as tendons and ligaments (*Bland & Ashhurst, 1997*; *Clark & Stechschulte, 1998*) and perhaps due to other factors not yet understood. This uncertainty about the causes of variability in the patella may also relate to incomplete understanding of its mechanical loading and function in vivo, as follows.

Where a patella is present in its typical form, its primary function is to modify the mechanical advantage (ratio of output force to muscle force) at the knee joint, by increasing the moment arm of the tendon in which it is embedded and thereby altering the amount of force needed from the quadriceps muscles in order to generate a particular moment (torque; rotational force) about the knee joint (*Alexander & Dimery, 1985*; *Fox, Wanivenhaus & Rodeo, 2012*; *Haines, 1974*; *Heegaard et al., 1995*; *Herzmark, 1938*; *Howale & Patel, 2013*; *Tecklenburg et al., 2006*). In humans, the patella causes the quadriceps muscle group's moment arm about the knee to increase as the knee becomes more extended, causing the amount of quadriceps muscle force required per unit of patellar tendon force (i.e. at the insertion onto the tibial tubercle) to vary significantly across knee joint flexion–extension (*Aglietti & Menchetti, 1995*; *Fellows et al., 2005*). By articulating with the femur, the patella also transmits some forces of the quadriceps muscle group directly onto the femur (the patellofemoral joint reaction force); forces which can reach a maximum of 20–25 times body weight (*Aglietti & Menchetti, 1995*).

The mobility of the patella is an important aspect of its function. While, in humans, the patella mostly flexes and extends relative to the femur as the knee is flexed and extended, it also translates and pitches (tilts) and rolls (*Aglietti & Menchetti, 1995*; *Fellows et al., 2005*), leading to variable contact between the patella and femur that is reflected in the angled facets of the human patella (*Lovejoy, 2007*). In contrast to the situation in humans (as well as in early hominins such as *Australopithecus*), in chimpanzees and presumably many other primates (as well as other taxa such as sheep *Bertollo, Pelletier & Walsh, 2012*, *2013*), the patella remains in tight articulation with the femur throughout the knee's range of motion, reducing patellofemoral stresses especially when the knee is strongly flexed, as it habitually is in those non-human primates (*Lovejoy, 2007*). Other primates show varying degrees of specialization of patellar morphology that alter the moment arm of the patellar tendon, with great apes apparently having a patella most specialized for widely varying knee joint postures (*Pina et al., 2014*). It has been claimed that in hominids and ursids (bears) alike, there is an association between plantigrady (flat-footedness), increased knee range of motion and patellar mechanics (*Lovejoy, 2007*); that is an interesting hypothesis that could be rigorously tested.

In the elbow of humans and other mammals, there is an extension of the ulna called the olecranon (process), which serves a lever-like function analogous to that of the patella (*Herzmark, 1938*). However, a mobile sesamoid bone-like the patella has a more flexible ("dynamic gearing") function in improving mechanical advantage compared with an immobile retroarticular process-like the olecranon (*Alexander & Dimery, 1985*). There tends to be an inverse relationship between mechanical advantage and speed of joint motion (*Hildebrand, 1998*), thus a high mechanical advantage is not necessarily useful in all cases, which may in part explain the variable occurrence, size and shape of the patella in animals with different lifestyles and modes of locomotion. Biomechanical studies of primates (*Lovejoy, 2007*; *Pina et al., 2014*) and domesticated mammalian species (e.g. dogs *Griffith et al., 2007*; *Kaiser et al., 2001*, sheep *Bertollo, Pelletier & Walsh, 2012*, *2013*, horses *Schuurman, Kersten & Weijs, 2003*; *Wentink, 1978*) have contributed some knowledge of how the patella functions in these groups, or in individual species, but a general "functional synthesis" for the patella is still lacking.

*De Vriese (1909)* performed pioneering comparative analyses and attempted syntheses of patellar size and morphology in comparison to other leg bones, between species and among multiple individuals in some species. No clear correlations were observed between the size of the patella and other major hindlimb bones (femur, tibia and fibula). A correlation was claimed between the sizes of the patella and the talus (or intermedium) in the ankle, although no clear, plausible mechanistic/functional justification was suggested and no statistical analyses were performed. Somewhat oddly, no relationship was evident between the size and shape of the patella and the femoral patellar groove (*De Vriese, 1909*). The more restricted but quantitative analysis of *Valois (1917)* focused mainly on primates and challenged many of De Vriese's claims that mechanical or physiological explanations of patellar morphology have "no scientific merit". *Haxton (1944)* also criticized De Vriese for focusing on relative length of bones; his own "patellar index" based on relative width found no correlation with animal speed or size, but he

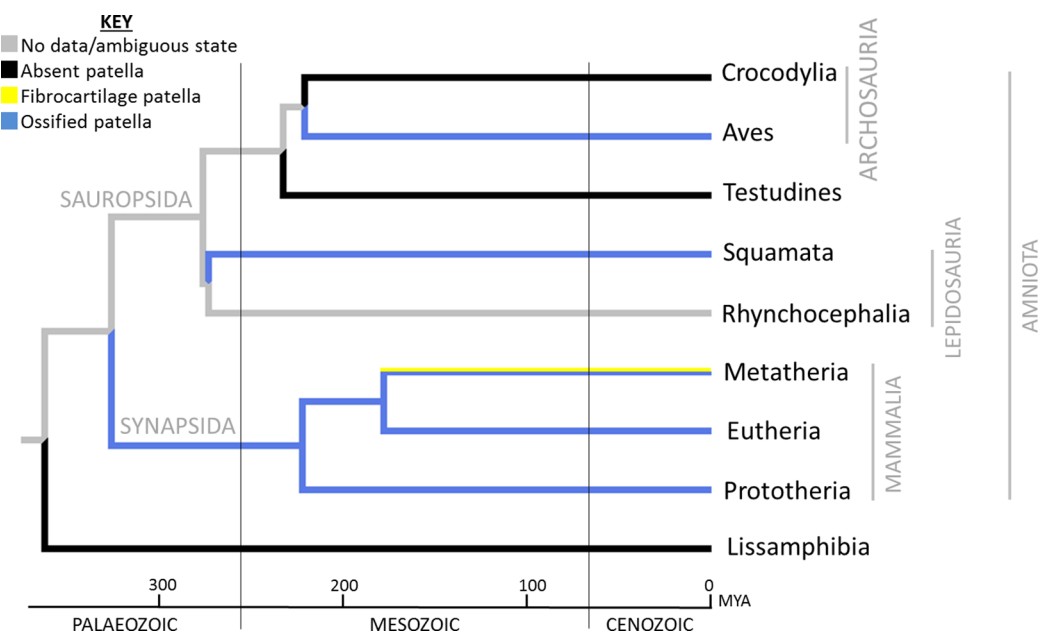

**Figure 3** **Reconstruction of ancestral patellar states in Tetrapoda, showing the major extant clades.**
Reconstruction was performed using Mesquite's parsimony algorithm and unordered character states,
where 0 (black) = absent patella, 1 (yellow) = soft tissue patella/patelloid and 2 (blue) = ossified patella;
see "Materials and Methods" for further details. The distribution of the ossified patella among extant
clades has been interpreted as three occasions of independent evolution (in Aves, Squamata and
Mammalia) (*Dye, 1987*; *Haines, 1940*), a conclusion strongly reinforced by specific fossil evidence
(absence or equivocality of a patella in all outgroups). Reconstruction within Mammalia is explored in
more depth in Figs. 5–7. Mya, millions of years from present.

inferred that the patella confers functional advantages in knee extension. There has been
little examination of these questions in a modern comparative, rigorously statistical or
biomechanical context since these studies. A notable exception is a study of the distal
femur and patellar groove in bovid mammals, indicating increased mechanical advantage
of the knee in larger species (*Kappelman, 1988*).

The occurrence of an ossified patella in the knee joint is not universal among tetrapods
(Fig. 3). A bony patella is absent in extinct early Tetrapoda and crown clade Lissamphibia
(*Dye, 1987*; *Haines, 1942*; *Herzmark, 1938*; *Vickaryous & Olson, 2007*), all non-avian
dinosaurs, Crocodylia, and Testudines (turtles), and all other extinct tetrapods.
*Hebling et al. (2014*; their Fig. 3A*)* illustrate what seems to be a patella formed of soft
tissue in the bullfrog *Lithobates catesbeianus*. That fascinating observation needs a more
comprehensive examination across Anura and Urodela to test if a soft tissue "patelloid" is
ancestral for Lissamphibia or smaller clades. In contrast, an ossified patella is present
in many or most Squamata (lizards and kin) with limbs (*Camp, 1923*; *Carrano, 2000*;
*De Vriese, 1909*; *Dye, 1987*, *2003*; *Gauthier et al., 2012*; *Haines, 1940*, *1942*; *Hutchinson,
2002*, *2004*; *Jerez & Tarazona, 2009*; *Maisano, 2002a*; *Regnault et al., 2016*; *Vickaryous &
Olson, 2007*). Patellar status (used throughout our study to refer to presence/absence
of ossification in adults) is unknown for the (mostly extinct) Rhynchocephalia (sister
group to Squamata), although a patella is at least sometimes present in the tuatara

*Sphenodon*—the only extant rhynchocephalian (*Regnault et al., 2016*). An apparent sesamoid bone was noted in the knee joint region of a specimen of *Macrocnemus*, a mid-Triassic (~235 Mya) reptile, which may be the earliest identified occurrence of a patella in any animal group (*Rieppel, 1989*), although this structure may have been a different sesamoid bone or ossicle. There have been anecdotal accounts of fibrocartilaginous or "fibrovesicular" patelloids in some reptiles such as turtles and crocodiles (*Haines, 1940*, *1942*; *Pearson & Davin, 1921a*, *1921b*), but these are not well explored. Thus, although such fibrous tissues seem to be excellent candidates for intermediate evolutionary character states between "absence of ossified patella (normal extensor tendon)" and "presence of ossified patella", empirical grounding for this transformational sequence within Sauropsida is weak.

No patella has been observed in early, stem-group birds throughout the Jurassic and Cretaceous periods, except in the well-documented Cretaceous Hesperornithes, diving birds with vestigial wings and an extremely large and unusually shaped patella, resembling that in some extant diving birds (*Lucas, 1903*; *Marsh, 1875*; *Martin, 1984*; *Martin & Tate, 1976*; *Shufeldt, 1884*; *Thompson, 1890*). A patella is found in some Cenozoic fossil bird specimens, most notably archaic penguins and commonly among many crown clade birds (*Dye, 1987*, *2003*; *Hutchinson, 2001*, *2002*; *Ksepka et al., 2012*; *Shufeldt, 1884*; *Vickaryous & Olson, 2007*; *Walsh & Suárez, 2006*). Our recent study (*Regnault, Pitsillides & Hutchinson, 2014*) inferred that a patella was probably ancestrally present in the common ancestor of Hesperornithes and living birds over 70 Mya. However, the bony patella was lost (and in some cases replaced by fatty cartilaginous tissue) in some large flightless birds such as emus, cassowaries and the extinct moa, yet unexpectedly is present as a double ossification in the knee joints of ostriches (*Chadwick et al., 2014*).

An osseous patella is generally found in two of the three crown groups of Mammalia: Eutheria (Fig. 3) and Monotremata (see Figs. 4A–4D), but not in most Metatheria (see Figs. 4E and 4F) (*Dye, 1987*, *2003*; *Vickaryous & Olson, 2007*). This raises the question whether this patella represents independent, convergent evolutionary origins in the Eutheria and Monotremata, or an ancestral origin for all three groups, with loss of the ossified patella amongst most Metatheria. To address this question, we conducted phylogenetic character mapping with Mesquite software (*Maddison & Maddison, 2017*) that reconstructed patellar evolution in Mammalia. Using likelihood methods, we also traced the most likely pattern of evolution over existing phylogenies, and considered alternate proposed topologies to test how they affected our reconstructions. Based on the predicted evolutionary patterns and individual morphologies, we propose suggestions as to the lifestyle of particular taxa, and consider where general correlations between lifestyle and patellar presence/absence might exist (or not).

*Mottershead (1988)* called the patella "that prince among sesamoids" but questioned whether it is "not typical of its kind". But is there even a "typical" patella (bony or otherwise)? Our synthesis of key data from morphology and function to phylogeny, development and genetics allows us to evaluate just how "typical" any patella is, even for a mammalian patella.

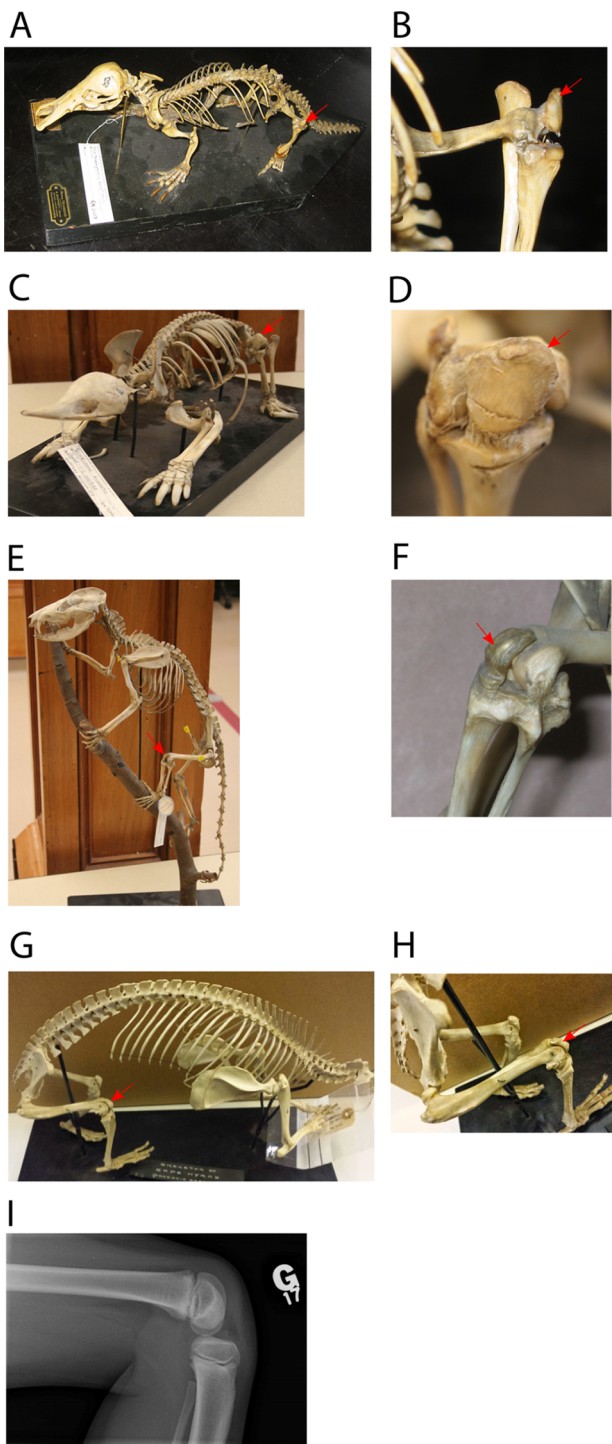

**Figure 4 Examples of tetrapods with or without patellae.** Red arrows denote the patella. (A, B) *Ornithorhynchus anatinus* (Monotremata: duck-billed platypus, Redpath Museum specimen 2458). (C, D) *Tachyglossus aculeatus* (Monotremata: echidna, Redpath Museum specimen 2463). (E, F) *D. virginiana* (Metatheria: North American opossum, Redpath Museum specimen 5019). (G, H) *Procavia capensis* (Eutheria: Afrotheria: Cape hyrax, uncatalogued Horniman Museum Specimen, London, UK). (I) knee of patient with Meier–Gorlin Syndrome (*Guernsey et al., 2010*). For more images of mammalian patellae (or lack thereof in some marsupials), see Figs. S1–S3.

## MATERIALS AND METHODS

Our methods followed standard phylogenetic character mapping (i.e. evolutionary reconstructions) methods in comparative biology (e.g. *Baum & Smith, 2013*; *Cunningham, Omland & Oakley, 1998*; *Huelsenbeck, Nielsen & Bollback, 2003*); with details as follow. We surveyed the literature and additional specimens (Fig. 4; Table S1; Figs. S1–S3) and coded the patella as absent (score = 0), fibrocartilaginous (i.e. "patelloid"; score = 1), or ossified (score = 2) for each taxon in our analysis, with "?" denoting an ambiguous character coding. We did not code the "suprapatella" here, as there is substantial confusion over its homology. We used two phylogenetic optimization methods in Mesquite software (*Maddison & Maddison, 2017*) to reconstruct possible evolutionary polarity of the patella in the clade Mammaliamorpha (with a focus on Mammaliaformes), as follows. First, for broad reconstruction across Tetrapoda, we used a phylogeny based on *Gauthier, Estes & De Queiroz (1988)* and *Shedlock & Edwards (2009)*, with average branch lengths they derived from several studies. Some aspects of the phylogeny remain controversial, such as the position of Testudines (turtles; *Hedges, 2012*). Reconstruction was performed using Mesquite's parsimony algorithm and unordered character states and results are illustrated in Fig. 3. As this analysis only involved major clades and not any stem lineages, it was intended as purely illustrative of general patterns and the current state of knowledge, given that patellar evolution across Tetrapoda had not been analysed phylogenetically before.

We adopted composite phylogenetic trees for our study taxa (*Archibald, 1998*; *Beck, 2012*; *Bi et al., 2014*; *Cardillo et al., 2004*; *Forasiepi et al., 2006*; *Gatesy et al., 2013*; *Goloboff et al., 2009*; *Kielan-Jaworowska, Cifelli & Luo, 2004*; *Luo, Kielan-Jaworowska & Cifelli, 2002*; *Luo et al., 2003*; *Luo, 2007a, 2007b*; *May-Collado, Kilpatrick & Agnarsson, 2015*; *Meredith et al., 2009*; *Meredith et al., 2011*; *Mitchell et al., 2014*; *O'Leary et al., 2013*; *O'Leary & Gatesy, 2008*; *dos Reis et al., 2012*; *Rose, 2006*; *Sánchez-Villagra et al., 2007*; *Song et al., 2012*; *Spaulding, O'Leary & Gatesy, 2009*; *Springer et al., 2003, 2007*; *Springer, Krajewski & Meredith, 2009*; *Thewissen, 1990*; *Thewissen et al., 2007*; *Wible et al., 2007*; *Zack et al., 2005*). As defined by several authors, the clade Mammaliaformes includes crown group Mammalia plus closely related extinct stem-mammals such as the iconic *Morganucodon* and the more recently discovered *Sinoconodon*, and is characterized by diagnostic features involving the teeth, jaw and inner ear (*Kielan-Jaworowska, Cifelli & Luo, 2004*; *Rose, 2006*). Extant mammals (crown group Mammalia) include three main clades: Placentalia, Marsupialia and Monotremata. Placentalia lie within the Eutheria, Marsupialia lie within the Metatheria and Monotremata lie within the Australosphenida, all of which diverged during the Mesozoic, pre-dating the K–Pg extinction event ~66 Mya.

The overall phylogeny used for Mesozoic mammals (Fig. 5) was based on the topology of *Bi et al. (2014*; their main Fig. 4 and extended data Fig. 9*)*. However, we chose to show *Henkelotherium* branching prior to *Vincelestes* following (*Luo, 2007b*) because their relationship with Theria was less well-resolved in *Bi et al. (2014)*. Approximate divergence times for key clades were taken from *Bi et al. (2014*; Fig. 4*)*. Divergence of *Vincelestes*, *Henkelotherium* and *Akidolestes* came from *Luo (2007b)*. The remaining undated

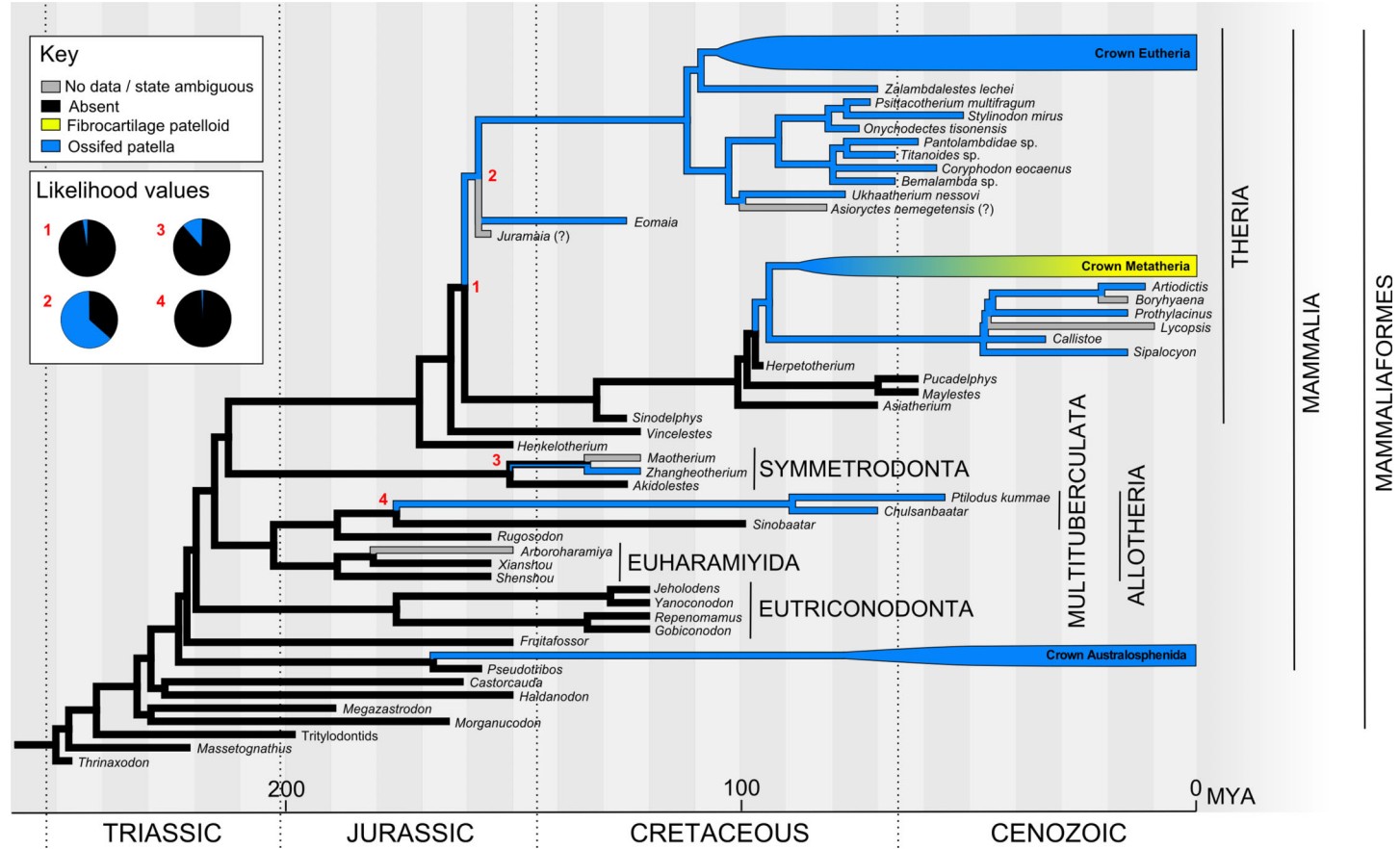

**Figure 5 Ancestral state reconstruction of the patella in Mesozoic mammals (see Fig. S4 for alternative tree topology).** The main tree shows a parsimony reconstruction using unordered character states, where branch colour indicates reconstructed state. Maximum likelihood gives similar results to parsimony, and likelihood values for numbered nodes are displayed (inset). Crown Metatheria and Eutheria are further explored in Figs. 6 and 7. Our results suggest that the ossified patella has evolved at least five times within Mammaliaformes.

divergences and branch lengths were estimated using data from the Palaeobiology database (http://www.fossilworks.org/), accounting for the date ranges of fossil taxa.

The topology of the metatherian tree was based on several sources that are all fairly congruent with one another. *Sinodelphys* was least nested, as in *Luo et al. (2003)*, followed by *Asiatherium*, *Pucadelphys* + *Mayulestes*, *Herpetotherium* and crown Marsupalia as shown by *Sánchez-Villagra et al. (2007)* also by *Beck (2012)* and *Luo et al. (2003)*. Sparassodonta were sister to crown Marsupialia (*Babot, Powell & de Muizon, 2002*; *Forasiepi et al., 2006*; *Suarez et al., 2016*). The topology and divergence dates of crown Marsupialia were from *Mitchell et al. (2014)*. Divergence dates of *Sinodelphys*, *Asiatherium* and of *Pucadelphys* from *Mayulestes* were from *Luo et al. (2003)*. Dates within Sparassodonta were taken from *Forasiepi (2009)*. The remaining undated nodes were estimated, so that the interbranch lengths between dated nodes was approximately equal.

The topology of basal eutherians used *Hu et al.'s (2010)*, with *Juramaia* polytomous with *Eomaia* and crown Placentalia as in *Luo et al. (2011)*, which also brought the basal eutherian node back to ~160 Mya. Alternative placement of *Eomaia* as a stem therian as in

*O'Leary et al. (2013)* was also explored as a supplementary analysis. The branch order of the main crown Placentalia clades (Xenarthra, Afrotheria, Euarchontoglires and Laurasiatheria), as well as the placement of many of the extant and fossil groups, came from *O'Leary et al. (2013)*. Divergence dates of extant taxa were estimated from the Timetree database (http://www.timetree.org; *Hedges, Dudley & Kumar, 2006*). Divergence dates of fossil taxa were from *O'Leary et al. (2013)* or estimated from fossil dates from the Palaeobiology database as above.

Exceptions and expansions to the topology of *O'Leary et al. (2013)* were as follows: (1) the placement of Pantodonta and Taeniodonta is ambiguous, but both groups were suggested to be derived from the cimolestids (*McKenna & Bell, 1997*). Here, we placed these groups as stem eutherians (*Rook & Hunter, 2014*). (2) Within primates, we placed *Omomys, Teilhardina, Archicebus, Notharctus* and *Plesiadapis* (*Ni et al., 2013*). (3) Within Glires, *Nonanomalurus* was classified with Anomaluroidea, diverging from the group containing Sciuridae (*Marivaux et al., 2016*), and adopting a divergence date of 60 Mya. Apatemyids like *Apatemys chardini* may be basal members of Euarchontoglires, with weak support for a sister-group relationship with Glires (*Silcox et al., 2010*). (4) The topology within Carnivora was based on *Flynn et al. (2005)*. (5) The detailed topology within Cetartiodactyla followed *Spaulding, O'Leary & Gatesy (2009)*. *Maiacetus* was placed alongside *Rodhocetus* and *Artiocetus* (within Protocetidae). *Gervachoerus* was placed tentatively alongside *Diacodexis* (as it is classified within Dichobunoidea); its actual placement is unclear. *Paratylopus, Merychyus* and *Protoreodon* were placed near to *Camelus*, within Camelidamorpha, but again their exact relationships are unclear. (6) The detailed topology of Perissodactyla followed *Holbrook & Lapergola (2011)*. Notoungulata and *Eoauchenia* (Litopterna) were placed sister to Perissodactyla (*Welker et al., 2015*). Following recent analyses (e.g. *Cooper et al., 2014*), we placed Phenacodontidae and Desmostylia as stem perissodactyls. (7) The position of Dinocerata is controversial. Here, we placed Dinocerata within Laurasiatheria, close to Perissodactyla and Cetartiodactyla (*Burger, 2015*), until more data on the placement of this group become available. (8) The detailed topology within Chiroptera followed *Simmons et al. (2008)*.

Our analysis involved numerous challenges and caveats. Many anatomical studies of extant or extinct species omit any mention of the patella, leaving its provenance in these taxa as uncertain. Interpretation of patellar status is especially challenging in fossils due to the rarity of finds with extensive, articulated postcranial material, the potential occurrence of other small non-patellar bones in the knee joint, and the uncertain age of the animal at time of death versus the developmental timing of sesamoid ossification (usually unknown; often relatively late in ontogeny). For the present analysis, statements in the primary literature regarding patellar status were generally accepted at face value except when superseded by more recent observations. From some publications with high quality photographs, patellar status was tentatively interpreted even if not discussed in the original text. In some cases, patellar status was confirmed by direct observation (e.g. Fig. 4; Figs. S1–S3; Table S1). Drawings found in secondary citations were mostly not been taken as definitive evidence, as we noticed examples of discrepancies between primary references and such drawings found in review articles or even textbooks, which may

simply assume patellar status in mammals. Also many mammalian groups are found over long temporal and geological spans, thus we were cautious about using the presence of a patella in one or a few individual extant or fossil specimens to infer presence throughout the group, although in some cases there was clearly enough conservatism within a clade to score it for all members.

An important knee structure related to the patella is the femoral patellar or intercondylar sulcus (groove) (*Norell & Clarke, 2001*; *Polly, 2007*). This sulcus is anatomically associated with a true patella (Figs. 1 and 4) in terms of its direct role in guiding the patellar sesamoid and tendon's path of movement during leg flexion/extension, and in mediolaterally confining the patellar tendon, which may enhance osteogenic stresses favouring the formation of a patella (*Sarin & Carter, 2000*; *Wren, Beaupre & Carter, 2000*). In the absence of an observed patella in fossil specimens, this sulcus at the distal end of the femur is sometimes treated as evidence of a patella even in the absence of the observed bone itself. We deemed this conclusion to be unwarranted. For example, the evolution of a patellar sulcus in early pygostylian birds substantially predated the evolution of an ossified patella in later ornithurine birds; moreover, the sulcus was retained in some avian taxa that lost the patella (*Clarke & Norell, 2002*; *Hutchinson, 2002*; *Livezey & Zusi, 2006*; *Regnault, Pitsillides & Hutchinson, 2014*). In contrast, a prominent sulcus is absent in many Squamata despite the presence of a patella (S. Regnault & J. R. Hutchinson, 2016–2017, personal observation). Together these observations indicate that these two anatomical features are not obligatorily coupled, so reliance on the observed presence of an ossified patella in fossil specimens was warranted. Nonetheless, at least among mammals the complete absence of a femoral patellar sulcus might be indicative of the absence of an ossified patella (*Chester et al., 2012*).

## RESULTS AND DISCUSSION

Our overall evolutionary reconstruction of the patella for Mesozoic mammals is shown in Fig. 5, for Metatheria/Marsupialia in Fig. 6, and for Cenozoic Eutheria/Placentalia in Fig. 7, with details for specific taxa in Table S1 and alternative phylogenetic analyses in Figs. S4 and S5. Here, we sequentially summarize and discuss our findings for five subgroups of Mammaliamorpha (especially Mammaliaformes): (1) Mesozoic pre-therians and stem-therians; (2) Mesozoic Metatheria and Eutheria; (3) Cenozoic Monotremata; (4) Cenozoic Metatheria and (5) Cenozoic Eutheria. We then conclude with a general synthesis of our study's insights (as well as uncertainties) and a consideration of how available and emerging data on developmental genetics of the patella might help shed light on the "evo-devo" of the patella, augmenting the phylogenetic and anatomical insights that this study focuses on.

### Mesozoic pre-therian and stem-therian mammals

The earliest mammals as widely construed include *Sinoconodon*, the Morganucodonta and Docodonta. These were mostly small, probably insectivorous animals, that appear to have lacked a patella, although it is unclear whether the known specimens contain sufficient postcranial material or are from verified adults, to allow for definitive conclusions.

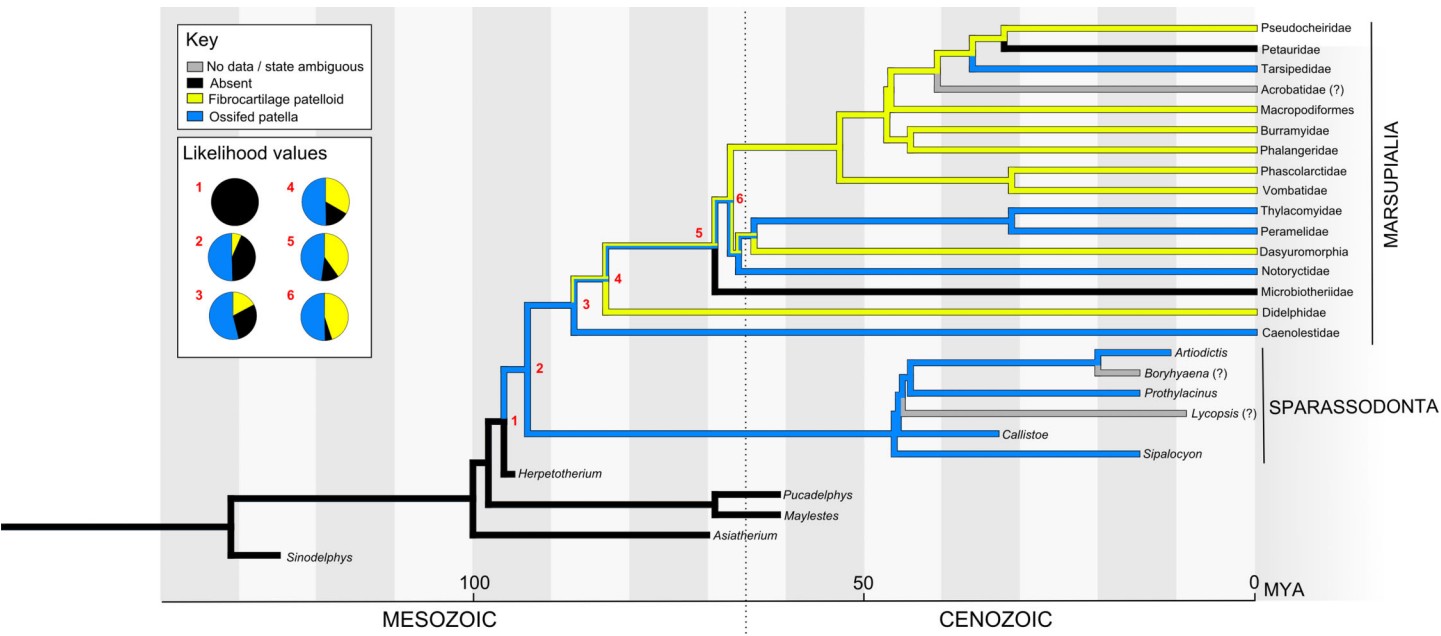

**Figure 6 Ancestral state reconstruction of the patella in Metatheria and related taxa.** The main tree shows a parsimony reconstruction using unordered character states, where branch colour indicates reconstructed state. Likelihood values for the numbered nodes are shown (inset). Our results suggest that the ossified patella evolved once in Metatheria, with instances of loss and reversion (to a fibrocartilaginous patelloid and back).

The absence of a clear patella in two stunningly preserved docodonts (the scansorial [climbing-adapted] *Agilodocodon* and fossorial [digging-adapted] *Docofossor*) lends credence to the conclusion that it was generally absent in early mammaliaforms (*Luo et al., 2015b*; *Meng et al., 2015*). There is convincingly strong evidence of absence of a bony patella in earlier pre-mammals in lineages dating from the divergence of Synapsida and Sauropsida/Reptilia (~320 Mya), including the early "pelycosaurs", therapsids and cynodonts (*Kemp, 2005*).

Australosphenida, the clade containing and thus ancestral to extant Monotremata, diverged from other mammals extremely early, possibly in the mid-Jurassic (*Kielan-Jaworowska, Cifelli & Luo, 2004*). There is little postcranial material for any extinct members of this lineage however, and no hindlimbs (*Kemp, 2005*). The patella in crown clade monotremes is discussed below.

*Fruitafossor,* from the late Jurassic (150 Mya), diverged after the Australosphenida (*Luo & Wible, 2005*). Its relationship to other early mammals is complicated by its mixture of characters in the molar teeth, middle ear and elsewhere. *Fruitafossor* is described as lacking a patella, and it is proposed to have had a fossorial lifestyle.

The Eutriconodonta were found abundantly across the world from the middle Jurassic to early Cretaceous periods (*Kielan-Jaworowska, Cifelli & Luo, 2004*). Among eutriconodonts, a poorly developed patellar groove on the distal femur is found but an ossified patella is absent.

The Allotheria were an extremely successful and widely dispersed group of mammals, among which the best understood are the multituberculates (*Kielan-Jaworowska, Cifelli &*

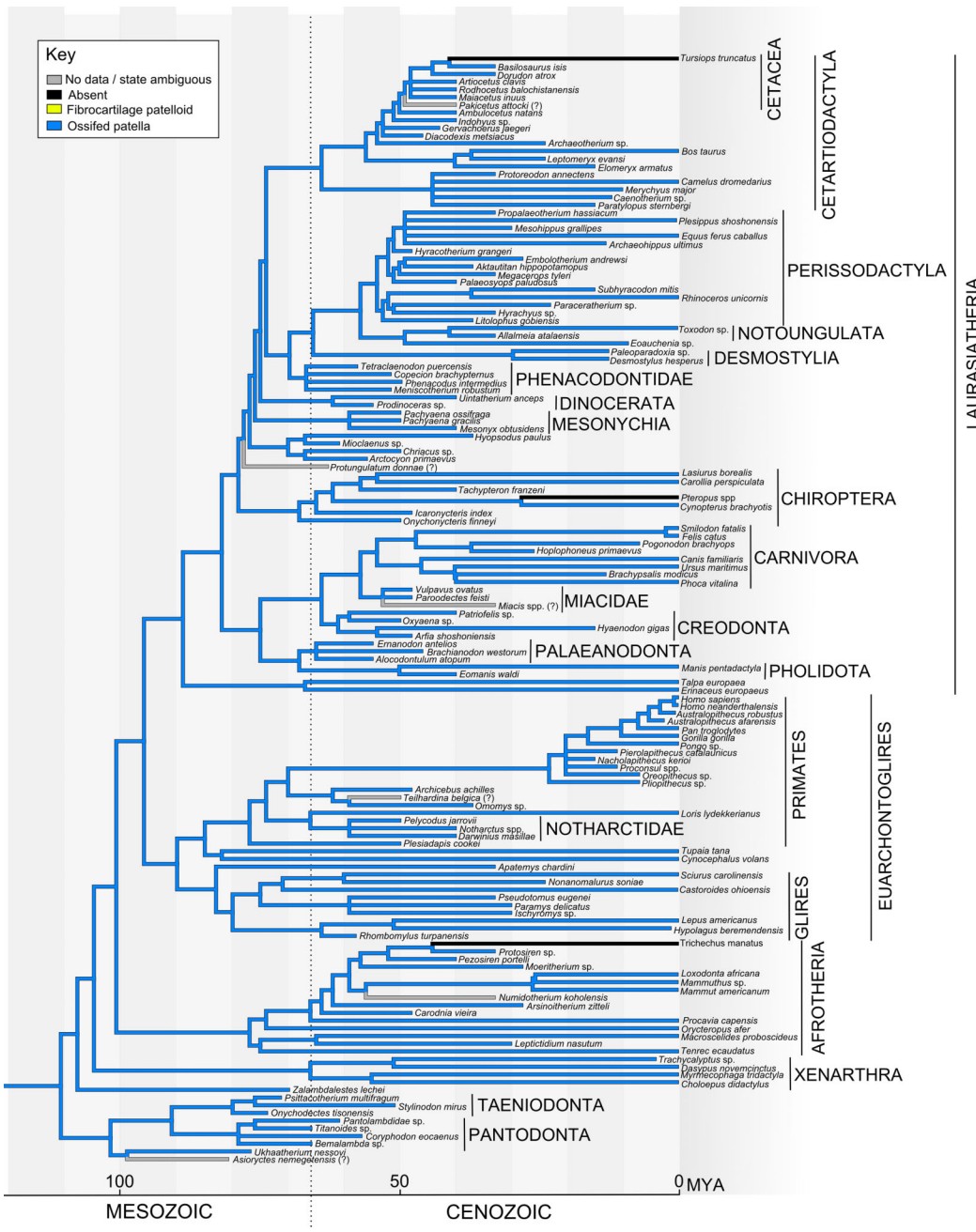

**Figure 7** **Ancestral state reconstruction of the patella in Eutheria.** The main tree shows a parsimony reconstruction using unordered character states, where branch colour indicates the reconstructed state. Our results suggest that the ossified patella evolved only once within Eutheria and (as far as is currently known) has only been lost by the bat genus *Pteropus* (not counting groups which have lost hindlimbs, e.g. *Trichechus manatus*/crown Sirenia, *Tursiops truncatus*/crown Cetacea).

*Luo, 2004*; *Wilson et al., 2012*). Generally Allotheria are found from the late Triassic to the Eocene; thus this group spanned the heyday of the non-avian dinosaurs and survived the K–Pg extinction (*Kielan-Jaworowska, Cifelli & Luo, 2004*). Multituberculates were predominantly small animals, either herbivorous or omnivorous (*Kielan-Jaworowska,*

*Cifelli & Luo, 2004*). A patella is noted for the nearly complete multituberculate *Ptilodus*, a proposed scansorial animal from the early Cenozoic. A patella is also present in the Cretaceous multituberculate *Chulsanbaatar*. It is unclear whether a patella is typical of all members of the multituberculate group and is under-reported due to lack of hindlimb material for most group members, or whether it occurs only among selected species, although the former seems more plausible. A patella is not reported, however, for the early Jurassic *Rugosodon*, a proposed multituberculate specimen with one relatively intact knee joint (*Yuan et al., 2013*), so it is conceivable that an ossified patella evolved later within the Allotheria (Fig. 5).

Specimens of the diverse group "Haramiyida" are mostly restricted to cranial material, and the relationship of this ancient group to other Allotheria and Mammaliaformes has been controversial (*Butler, 2000*; *Kielan-Jaworowska, Cifelli & Luo, 2004*; *Rose, 2006*). However, several recently described more complete haramiyid specimens from the Jurassic with at least one preserved knee joint lack a patella (*Bi et al., 2014*; *Zheng et al., 2013*; *Zhou et al., 2013*). These new specimens have been interpreted to support an Allotheria clade including a paraphyletic "Haramiyida" (but a valid clade Euharamiyida including many "haramiyid" taxa) and Multituberculata (Fig. 5), although new analyses of a key specimen of *Haramiyavia* concluded that the haramiyids and multituberculates were not closely related (*Luo et al., 2015a*). The inclusion of the "Euharamiyida" in Allotheria pushes the divergence date of the group significantly earlier into the late Triassic, whereas multituberculates themselves appear only in the middle to late Jurassic. Final resolution of this controversy will undoubtedly require additional fossil material.

Symmetrodonta were a group of diverse, small mammals widely distributed in time from the late Triassic to the late Cretaceous (*Kielan-Jaworowska, Cifelli & Luo, 2004*). In the subgroup of spalacotheroids, a patella is reported for one fairly complete specimen (*Zhangheotherium*) but not for another (*Akidolestes*) (*Chen & Luo, 2012*; *Luo & Ji, 2005*) (these two specimens are coded oppositely in character matrices in some subsequent publications *Bi et al., 2014*; *Zhou et al., 2013*, probably in error); a patella seems absent in *Maotherium*.

Eupantotheria was a diverse group found commonly from the mid-Jurassic to the early Cretaceous (*Kielan-Jaworowska, Cifelli & Luo, 2004*). The patella is reported as absent in both an early European specimen (*Henkelotherium*, late Jurassic) and a later South American specimen (*Vincelestes*, early Cretaceous) (Fig. 5). The large group of dryolestid Eupantotheria possibly survived past the K–Pg boundary, and have an unknown patellar status.

The tribotherians were the earliest-diverging group to share key molar features with the therians. However, no postcranial specimens have been reported; thus, nothing is known of their patellar morphology (*Kielan-Jaworowska, Cifelli & Luo, 2004*).

The single specimen of *Juramaia* from the Jurassic (~160 Mya) unfortunately lacks hindlimb material; therefore, its patellar status is unknown. Based on its forelimb, *Juramaia* is proposed to have been scansorial or possibly arboreal (*Luo et al., 2011*). The later specimen of *Eomaia* from the early Cretaceous includes all limb elements, and is described with a patella (*Ji et al., 2002*). Based on limb and foot features, *Eomaia* was

probably scansorial or arboreal. In the original publication, *Eomaia* was described as the earliest eutherian mammal (Fig. 5), however a more recent and much more extensive analysis confidently placed *Eomaia* prior to the eutherian/metatherian divergence (*O'Leary et al., 2013*) and thus at least as a stem member of the clade Theria (see Fig. S4). *Eomaia* (and presumably *Juramaia*) postdate the divergence of the Symmetrodonta, but their positions relative to the Eupantotheria remain to be determined, as does any close relationship between these two key taxa. Lacking a better alternative, here we refer to these taxa as "Theria", and in Fig. 5 versus Fig. S4, consider the consequences of *Eomaia*'s phylogenetic position on our conclusions.

In surveying, the available data mapped onto our composite phylogeny (Fig. 5; Fig. S4), it becomes evident that an ossified patella evolved multiple times (at least four) along the mammalian stem lineages during the Mesozoic era, whether using parsimony or maximum likelihood optimisation methods: at some highly uncertain time in the long mammalian lineage that led to Monotremata, in multituberculates/Allotheria, in *Zhangheotherium* or a direct ancestor, and likely twice (or between one to three times, depending on the placement of *Eomaia*; see Fig. 5 and Fig. S4) in the clade containing *Eomaia* and Theria (Metatheria and Eutheria). This result remained the same if the Euharamiyida were not included with multituberculates but pre-dated crown Mammalia, as suggested by some recent studies (e.g. *Luo et al., 2015a*).

## Mesozoic Metatheria and Eutheria

The two major extant mammalian groups, the Metatheria and Eutheria (together forming the clade Theria), diverged as early as the Jurassic (Fig. 5). The earliest fossil identified as stem metatherian, *Sinodelphys*, dates from the early Cretaceous of China (125 Mya, approximately contemporary to *Eomaia*) and lacks a patella (*Luo et al., 2003*). A patella also seems absent in the less complete Cretaceous stem metatherian *Asiatherium* (*Szalay & Trofimov, 1996*).

The earliest known occurrences of the patella in definitive stem eutherians (Figs. 5 and 7) were in the late Cretaceous *Ukhaatherium* (*Horovitz, 2003*), a relatively unspecialized form, and in *Zalambdalestes* (*Wible, Rougier & Novacek, 2005*), a more specialized taxon sometimes described as resembling later lagomorphs (*Rose, 2006*). Patellar status at the crown group node for Theria (plus *Eomaia*) remains ambiguous (Figs. 5 and 6; Fig. S4), as we consider below.

## Cenozoic Monotremata

The origins of the Monotremata (egg-laying mammals) are poorly understood. They are considered extant members of the clade Australosphenida (the alternative term Prototheria has been superseded), and hence with early roots in the Mesozoic. Molecular studies based on the sequenced genome of the platypus corroborate the long held interpretation that the monotremes diverged prior to the metatherian/eutherian split, consistent with proposed fossil-based phylogenies (*Warren et al., 2008*). Unfortunately, there are almost no reported hindlimb specimens of any extinct monotreme (including probable early monotreme fossils found in South America; *Musser, 2003*), with the

exception of the Pleistocene *Zaglossus* (echidna) from Australia and New Guinea (which may be the same as the extant species of that name). Unfortunately, although fossil *Zaglossus* hindlimb elements exist, including an articulated knee, neither presence nor absence of the patella has been reported (*Murray, 1984*). The extant monotremes, the platypus (*Ornithorhynchus anatinus*) and the echidnas (Tachyglossidae, two genera *Zaglossus* and *Tachyglossus*; four known species) all have substantial patellae (see Figs. 4A–4D) (*Herzmark, 1938*; *Rowe, 1988*). It is unclear when the two extant monotreme genera diverged, although a date early in the Cretaceous has been proposed (*Rowe et al., 2008*), and it is impossible for now to date the appearance of the patella in the monotreme lineage. Regardless, an ossified patella is homologous for this crown clade (Fig. 5), and alternative phylogenetic topologies did not change the general pattern of patellar evolution (Fig. S4).

## Cenozoic Metatheria

All extant Metatheria are within the subgroup of Marsupialia, however non-marsupials did exist earlier during the Cenozoic. As documented in the pioneering study of sesamoids in Marsupialia by *Reese et al. (2001)*, an ossified patella seems to be absent in the great majority of extant marsupial species, both from Australia and the Americas (*Flores, 2009*; *Herzmark, 1938*; *Holladay et al., 1990*; *Reese et al., 2001*; *Rose, 2006*; *Rowe, 1988*), including the sole surviving North American marsupial, the opossum *Didelphis virginiana* (Figs. 4E and 4F). Many marsupials have other sesamoid bones in the knee region (e.g. the parafibula, lateral sesamoid or "sesamoid bone of Vesalli"; Fig. 1), as well as a fibrocartilaginous "patelloid", which may to some degree serve the mechanical function of a bony patella (*Reese et al., 2001*). However, the mechanics of a fibrous or bony patella remain essentially unstudied (to our knowledge) in non-placental mammals, so this is simply speculation. Studies have claimed some association between reduction of the patella in many marsupials and locomotor style or ecology (*Holladay et al., 1990*; *Reese et al., 2001*), but these deserve testing with more detailed sampling across phylogeny and ontogeny.

Nonetheless, an ossified patella is found in a small number of extant marsupial species among otherwise divergent clades, both from Australia: at least several Peramelidae or bandicoots, and the two marsupial mole species of *Notoryctes*); and from South America: *Tarsipes*, a honey possum; and several, and possibly all, Caenolestidae or shrew opossums (see Fig. 6: note collapse of several large clades in terms of total number of species, in which no species have been shown to possess a bony patella; Table S1).

Possibly uniquely among crown clade marsupials, bandicoots also possess a chorioallantois fused to the uterine epithelium (i.e. a true placenta) (*Freyer, Zeller & Renfree, 2003*; *Padykula & Taylor, 1976*), which combined with an osseous patella led to the initial suggestion that they might actually be eutherians (*Reese et al., 2001*). However, more recent molecular and fossil-based phylogenetic studies provide no support for that hypothesis of eutherian bandicoots (*Asher, Horovitz & Sanchez-Villagra, 2004*; *Meredith, Westerman & Springer, 2008*; *Sánchez-Villagra et al., 2007*; *Westerman et al., 2012*). Bandicoots clearly are metatherians, and their chorioallantois is thus a convergently

evolved trait rather than plesiomorphic. It remains to be determined whether an ossified patella is present in all or only some bandicoots, as so far it is only reported in the Peramelinae of dry or temperature forests of Australia, not yet in the Peroryctinae of tropical rainforests of New Guinea, or the more distantly related bilbies (*Groves & Flannery, 1990*; *Meredith, Westerman & Springer, 2008*; *Westerman et al., 2012*). Similarly, a comprehensive study of the Caenolestidae remains to be performed, much as a more thorough study of the major marsupial clade Diprotodontia (wombats, kangaroos and kin) is needed.

Not surprisingly given the absence of a bony patella in most extant marsupials, any evidence of a patella is absent in the early Cenozoic Metatheria *Pucadelphys, Mayulestes,* and the later *Herpetotherium.* Unexpectedly, a bony patella is reliably reported in the Borhyaenoidea, an unusual group of dog-like carnivorous South American marsupials found from the Palaeocene through the Miocene (*Argot, 2002*, *2003a*, *2003b*, *2003c*, *2004*; *Argot & Babot, 2011*; *de Muizon, Cifelli & Paz, 1997*). Patellar status in some members of Borhyaenoidea (e.g. *Borhyaena* itself and *Lycopsis Argot, 2004*), and in the more inclusive group Sparassodonta, is uncertain due to the incomplete state of specimens. *Szalay & Sargis (2001)* noted other enigmatic fossil patellae from the Palaeocene of Brazil that they assigned to Metatheria, but the phylogenetic relationships of those fragmentary remains are unclear and no patellae were shown. However, no ossified patella is reported in extant or recent carnivorous marsupials such as *Thylacinus.*

Two related, pernicious problems remain for interpreting the evolution of the patella in Metatheria that may have ramifications for all of Mammalia/Mammaliaformes. First, *Szalay & Sargis (2001*; pp. 164–165*)* reported the presence of an ossified patella in older individuals of *Didelphis virginiana* in their study of an ontogenetic series from this species. They stated (p. 165) "In older individuals there is occasionally an elongated and small sesamoid ossification within the tendon of the quadriceps femoris where it crosses the knee joint when the knee is flexed". However, this observation was not documented with illustrations or photographs (especially tissue histology or x-rays) and hence remains a tantalizing anecdote. Similarly, *Owen (1866)* commented that some marsupials had no ossifications in their patellar tendon but others had "only a few irregular specks of ossification" and a "distinct but small bony patella in the *Macropus Bennettii*". In contrast, *Reese et al. (2001)* and *Holladay et al. (1990)*, respectively sampled 61 specimens (~39 adults) from 30 species of marsupials and three macropodid specimens (of unknown maturity), documenting no ossified patellae except as noted in bandicoots, and their studies used clear methods for identifying ossified tissues. It remains possible that patellar ossification occurs variably in older individuals among Metatheria, which would help explain its patchy description in known taxa.

If the latter situation is the case (i.e. the literature is unclear about patellar ossification in marsupials because they have more inherent variability), then it relates to a second problem, a cladistic one of character coding and transformational homology (*sensu Brower & Schawaroch, 1996*; *Pinna, 1991*). Should character states of the patella in metatherians, or even all mammals and their kin, be coded as an ordered transformational series such as absent (0), fibrocartilaginous (1) or ossified (2), or as an unordered series

(i.e. should evolutionary steps be required to go from 0–1–2 as two steps, or unordered allowing 0–2 transformations as one step)? We chose the unordered character option by default for all crown group mammals, but where relevant explain how an ordered option changed (or did not change) our results. An endochondral ossification of the bony patella is certain, but a fibrocartilaginous or otherwise soft tissue composition of the patella (coded as state 1) in adults is not unambiguously the necessary (i.e. ordered) evolutionary precursor character state-to-state 2 (ossified patella in adults). The solution to both of these problems lies in more developmental data for the patella (bony and otherwise) in diverse mammalian species, in addition to more scrutiny of the adult morphology in extant and fossil Mammalia (especially Metatheria).

As noted briefly in the Introduction, many marsupials have a primarily fibrocartilaginous patelloid in place of an ossified patella and some other mammals may have a "suprapatella". The developmental and evolutionary relationships of these structures remain somewhat unclear, particularly as some marsupials with an ossified patella (e.g. bandicoots) also possess a patelloid (*Reese et al., 2001*), suggesting that the patelloid is not developmentally equivalent to the patella in marsupials (*Vickaryous & Olson, 2007*). If so, this would indicate independent evolutionary histories of these two structures. Further work is required to clarify the relationships of the patelloid and suprapatella at least in extant taxa, before definitive evolutionary trajectories can be inferred. We reiterate that, just because a patella-like structure is not ossified, that does not mean it is a distinct organ deserving a new name and different homology as a phylogenetic character—although it may be a distinct state of the character "patella". However, either of these two possibilities needs careful testing particularly for Metatheria.

A non-osseous patelloid/suprapatella is also found in several closely related modern placental clades that lie far from the base of Eutheria (Fig. 7), suggesting that these represent independent acquisitions. We have not attempted to explicitly reconstruct the evolution of the patelloid in Eutheria. *Lewis (1958)* and *Broome & Houghton (1989)* speculated that the mammalian patelloid might be a precursor to the tibial epiphysis (*Broome & Houghton, 1989*; *Lewis, 1958*)—a so-called "traction epiphysis" (*Vickaryous & Olson, 2007*). Yet considering that the patelloid evolved after the tibial tuberosity (and proximal tibial epiphysis as well as distal femoral epiphysis; *Carter, Mikić & Padian, 1998*) of mammals, not before it, and lies proximal rather than distal to the patella, we reject this hypothesis. More study of the evolution of mammaliaform long bone epiphyses, however, is warranted to strongly and more generally test for associations between any epiphyses and sesamoids. Furthermore, this same phylogenetic evidence indicates that the patelloid in Euarchontoglires, some Carnivora and bandicoots is not ancestrally associated with leaping or other behaviours (e.g. *Jungers, Jouffroy & Stern, 1980*). As *Walji & Fasana (1983)* caution, the ancestral mechanical environment of the patelloid/suprapatella and its roles in different behaviours remain unclear, although it does seem to be associated with knee hyperflexion, like a typical fibrocartilaginous "wrap-around" tendon (e.g. *Ralphs, Benjamin & Thornett, 1991*; *Alexander & Dimery, 1985*).

Our unordered parsimony reconstruction (Fig. 6) indicated that an ossified patella was absent in the ancestor of Metatheria, then evolved in the ancestor of Sparassodonta

and Marsupialia. The bony patella may have been lost in the basal lineages of Marsupialia (reconstructed state here was equally parsimonious between an ossified and fibrocartilaginous patella), with subsequent re-acquisition in certain groups (Tarsipedidae, possibly Notoryctidae, Thylacomyidae + Peramelidae and Tarsipedidae) (Fig. 6). Ordered parsimony reconstruction resulted in subtle differences; making some nodes less ambiguous (i.e. state 1 [patelloid present] within basal Marsupialia) and others more ambiguous (such as the ancestor of Sparassodonta and Marsupialia, which became equally parsimonious between states 1 and 2). In contrast, maximum likelihood reconstruction indicated a single origin of the osseous patella in Metatheria (Fig. 6), with reduction to a fibrocartilage patelloid (in Didelphidae and the clade containing Pseudocheiridae + Vombatidae) and re-acquisition of a bony patella (in Tarsipedidae) marginally more likely than multiple instances of ossified patella evolution. Because presence of a patelloid has not been clearly excluded in some extant marsupials (e.g. Petauridae, Acrobatidae) and is unlikely to be fossilized, its reconstruction must be treated carefully. Finally, alternative placement of Microbiotheriidae did not drastically alter our evolutionary reconstructions (Fig. S5), aside from making a single origin of the ossified patella slightly more likely. Overall, we caution that inferences about the evolutionary history of the patella in Metatheria must remain tentative until further data become available.

## Cenozoic Eutheria

The Placentalia include all extant Eutheria as well as some fossil stem taxa (Fig. 7). Although there is some fossil evidence for placentals pre-dating the K–Pg event (*Archibald et al., 2011*), as well as substantial molecular dating consistent with an older placental radiation, the timing of the placental radiation remains highly controversial. However, our major conclusions about patellar evolution in placentals are not dependent on how this controversy is ultimately resolved, as a recent large-scale phylogenetic analysis convincingly established the presence of an osseous patella as a derived character state in the ancestral placental irrespective of its true date of divergence (*O'Leary et al., 2013*).

Fossil evidence supports the presence of the bony patella in essentially all Cenozoic placental groups (Fig. 7; also see Table S1 and Figs. S1–S4, with citations therein). Specimens with sufficient hindlimb material to make a determination of patellar status are rare in the early Cenozoic Palaeogene period (~66–23 Mya), but Palaeocene groups in which an ossified patella has been reported include the Taeniodonta (small to medium sized fossorial animals), Pantodonta (early herbivores), Palaeanodonta (small, possible insectivores; perhaps related to pangolins), "Condylarthra" (a diverse assemblage of putatively related taxa, probably polyphyletic, including both herbivores and carnivores, many of which may be stem members of subclades within the placental crown group) and the Plesiadapiformes, a sister group to crown clade primates (and possibly members of the clade Primates as well) (*Bloch & Boyer, 2007*; *Silcox, 2007*). In general, the evolutionary relationships between Palaeocene taxa and more recent placentals remain enigmatic.

Eocene placentals include examples whose close relationships to modern groups are well accepted. Among Eocene groups (Fig. 7; Table S1), an osseous patella has been

reported in older, extinct groups such as "Condylarthra", Creodonta (carnivores), Mesonychia (carnivorous/omnivorous artiodactyls or cetartiodactyls), Dinocerata (large hippo/equid-like herbivores), Brontotheriidae (large rhino-like herbivores), and Notoungulata (diverse South American hoofed herbivores; probably related to Afrotheria) (*O'Leary et al., 2013*), as well as in extinct species (in parentheses, see Table S1 for citations) recognized as stem members of several extant groups: Glires (*Rhombomylus*), Perissodactyla (*Propalaotherium*), early Sirenia retaining hindlimbs (*Pesoziren, Protosiren*), Proboscidea (*Numidotherium, Moeritherium, Barytherium*), Rodentia (the horse-sized *Pseudotomus, Paramys*), Pholidota (*Eomanis*), Artiodactyla (*Gervachoerus*), early Cetacea retaining hindlimbs (*Maiacetus*) and Chiroptera (*Icaronycteris, Tachypteron*). A bony patella is also reported for several Eocene primates, including the lemur-like Notharctidae (*Northarctus*) and the tarsier-like *Omomys* and *Archicebus*, in addition to the enigmatic primate *Darwinius*.

Despite an extensive literature search, we found no reports attesting to the presence of an osseous patella in certain widely cited Palaeocene and Eocene species, including: *Protungulatum*, frequently cited as the earliest true placental; *Miacis, Vulpavus, Viverravus* and *Didymictis*, which were stem Carnivora (*Gregory, 1920*; *Heinrich & Houde, 2006*; *Heinrich & Rose, 1995*, *1997*; *Samuels, Meachen & Sakai, 2013*); *Pakicetus*, a fully quadrupedal early cetacean (though sometimes reconstructed with a bony patella as in Fig. 7 and Figs. S1M and S1N) (*Thewissen et al., 2001*); *Leptictis*, possibly related to crown clade lagomorphs (*Rose, 1999*); *Sinopa*, a creodont (*Matthew, 1906*); and the early primates *Adapis, Leptadapis, Teilhardina*, and *Cantius* (*Dagosto, 1983*; *Gebo et al., 2012*; *Gebo, Smith & Dagosto, 2012*; *Rose & Walker, 1985*; *Schlosser, 1887*; *Szalay, Tattersall & Decker, 1975*). There is no reason to expect that a bony patella is missing in these species. These absences are more likely due to incompleteness of the fossil record and/or literature descriptions and images. Moreover, the massive collections of Eocene specimens from the Messel and Green River lagerstätten in Germany and Wyoming have not yet been fully described (*Grande, 1984*; *Schaal & Ziegler, 1992*). There are many examples of an ossified patella in specimens from extant placental groups across the more recent Miocene, Oligocene, Pliocene and Pleistocene, but a comprehensive search of the literature for those geologic epochs was deemed redundant for our major conclusions.

Based on fossil/morphological evidence plus extensive genomic DNA sequencing, there is a consensus that crown clade placentals can be historically and geographically defined by four major groups: Xenarthra, Afrotheria, Euarchontoglires (further divided into Euarchonta; featuring Primates; and Glires) and Laurasiatheria (*Rose, 2006*). These in turn may be resolved, with somewhat less consensus, into 19 crown clade "orders" (Fig. 7) (*O'Leary et al., 2013*). In two of these orders, the afrotherian clade Sirenia and the cetacean branch of (Cet)artiodactyla (laurasiatherian clade), extant members have extensively reduced or absent hindlimbs and thus lack skeletal knee structures, including an osseous patella. In contrast, the bony patella is retained among the aquatic seals and sea lions in Carnivora, although unlike Sirenia and Cetacea these animals still display some terrestrial habits and thus presumably still employ the gearing mechanism that the patella

is involved in at the knee. An ossified patella is documented as present in at least some members of all other 17 placental "orders" (e.g. Figs. 4G, 4H and 7; Figs. S1–S3; Table S1) (*de Panafieu & Gries, 2007*; *De Vriese, 1909*; *Dye, 1987*; *Herzmark, 1938*; *Lessertisseur & Saban, 1867*; *Rose, 2006*).

The evolution of the Cetacea presents an interesting scenario regarding patellar evolution (Fig. 7). Cetaceans evolved from a common ancestor with other (cet) artiodactyls (*Spaulding, O'Leary & Gatesy, 2009*; *Thewissen et al., 2007*). Early artiodactyls (including cetaceans), such as *Diacodexis* and *Indohyus*, shared morphological similarities with both extant groups of Cetacea (toothed and baleen whales) and yet retained an osseous patella (*Rose, 1982*; *Thewissen et al., 2007*), much as stem Sirenia did (*Domning, 2001*; *Zalmout, 2008*). Patellar status in *Pakicetus*, a presumptive early cetacean with full hindlimbs, remains uncertain based on the primary literature, but presence is likely considering the presence of a bony patella in its closest relatives. *Rodhocetus* and *Ambulocetus*, probably semi-aquatic early cetaceans, still had large hindlimbs and ossified patellae (*Madar, Thewissen & Hussain, 2002*). The pelvis and hindlimbs are greatly reduced in the later cetaceans *Dorudon* and *Basilosaurus*, but a bony patella is still present in these animals (*Gingerich, Smith & Simons, 1990*; *Uhen, 2004*). It is not clear exactly when the patella was lost altogether in later cetaceans with increasingly reduced hindlimbs.

Bats present another interesting case of patellar evolution (Fig. 7; Table S1). An osseous patella is generally present in bats (*Pearson & Davin, 1921b*). A bony patella is also reported in a well-preserved hindlimb of an early Eocene bat, *Icaronycteris*, of intermediate form but proposed to be a microchiropteran (*Jepsen, 1966*). However, in studies of multiple genera of modern bats including members from both of the major subgroups Megachiroptera and Microchiroptera (which is possibly paraphyletic), a bony patella was noted as absent in four species of the megachiropteran *Pteropus* (flying foxes of various sizes), and a few individual species of *Cephalotes*, *Epomophorus* and *Vespertilio* (*De Vriese, 1909*; *Lessertisseur & Saban, 1867*; *Smith, Holladay & Smith, 1995*). No obvious lifestyle distinction was noted for the *Pteropus* genus as compared to many other bats, hence the loss of the ossified patella in members of this particular subgroup (and others) remains mysterious. In general, bat hindlimbs are highly derived, adapted to hanging and pulling rather than pushing. A few bats such as the vampire bats are actively quadrupedal (*Adams & Thibault, 2000*; *Riskin & Hermanson, 2005*). Bat hindlimbs are articulated in abduction, so that the knee faces dorsally; as in the original ancestral orientation for Tetrapoda (Fig. 2) (*Neuweiler, 2000*; *Schutt & Simmons, 2006*). There remains a need for a comprehensive study of the patella in bats (*Smith, Holladay & Smith, 1995* only studied 31 specimens of 13 species), but this is challenging due to the existence of >900 extant bat species (*Jones et al., 2002*). The microstructure of the "patelloid" in *Pteropus* is generally similar to that in many marsupials (e.g. deep layer of fibrocartilage; superficial layer of dense connective tissue contiguous with the quadriceps/patellar tendon) (*Smith, Holladay & Smith, 1995*). This also raises the question of whether the patella only ossifies later in adulthood in *Pteropus*, rather than not ossifying at all.

## General evolutionary patterns and ambiguities

Considering the above distributions of patellar presence/absence in Mammalia (Figs. 5–7; Figs. S4 and S5) and our data matrix (Table S1), the simplest interpretation of the evolutionary record of the patella in mammals (by parsimony and maximum likelihood mapping of presence/absence) is that this structure arose (i.e. ossified) independently at least four times (but possibly up to six), mostly during the Mesozoic era: (1) in Australosphenida ancestral to modern monotremes; (2) in Multituberculata (later than *Rugosodon*); (3) in Symmetrodonta (specifically in Spalacotheroidea that were ancestral to *Zhangheotherium* but not *Akidolestes*); (4–6) in early Theria (including Eutheria, Metatheria, *Eomaia* and related stem groups; depending on topology between one and three times in this clade). Conceivably, a single common patelloid precursor may pre-date the origins of the bony patellae, or the bony patella may have arisen fewer times and undergone loss (and re-gain) in some lineages, similarly to the pattern in Metatheria. Each of these scenarios remain difficult to test purely with fossil evidence, however, due to the typical lack of preservation of cartilaginous or fibrous structures.

Once the bony patella evolved in Eutheria, it was highly conservative in its presence (Fig. 7). There are very few examples of fossil or extant Eutheria in which the hindlimb remains intact but the patella is unossified in adults (e.g. *Pteropus*). A caveat is that many fossil specimens are not sufficiently complete for a definitive rejection of patellar ossification in those taxa. Still, the evolutionary stability of the osseous patella in Eutheria stands in contrast to its general variability across mammals, and suggests some conserved functional requirement and/or ontogenetic mechanism that remains to be determined.

Although an ossified patella is absent in the majority of Metatheria, it is reported in several groups (Fig. 6; Fig. S5). This likely represents some loss and regain(s) of the early metatherian bony patella. Importantly, in this case the presence of a fibrocartilaginous "patelloid" in most marsupials shows a clear evolutionary polarity from an ossified patella to a non-ossified patelloid, and back again in the case of the secondary gain of ossification, in each case within Metatheria (*Reese et al., 2001*). This "patella to patelloid" transition suggests the reverse may also be possible—that a soft tissue patelloid may represent the evolutionary precursor to an ossified patella—but it has yet to be clearly documented. There is no obvious lifestyle or biomechanical correlate among all four groups of osseous patella-bearing Metatheria: the notoryctid moles are underground burrowers, and bandicoots may dig for insects, but *Tarsipes* is a nectar feeder and the borhyaenoids/sparassodonts were largely terrestrial carnivores. In contrast, other Australasian carnivorous marsupials including the recently extinct thylacine, and the extant quoll, numbat and Tasmanian devil are not reported to have a bony patella.

The large size of the patella in the monotreme platypus might be related to its aquatic (and partly fossorial) lifestyle. The other monotremes, the echidnas, also burrow and the long-beaked species (*Zaglossus*) lives in underground dens—further suggesting an association between fossorial habits and the presence or enlargement of a bony patella in Monotremata, as well as in some fossil Mammaliaformes (multituberculates?) but

curiously not in other fossorial stem taxa (e.g. the docodont *Docofossor*). Reduction of the patella in the Cetacea and Sirenia is not intrinsically correlated with their aquatic lifestyle, but with the reduction of the hindlimbs as part of their particular adaptations. Elsewhere in groups with aquatic adaptations, for example in various diving birds, an unusually large patella is found. It seems premature to weave detailed scenarios around the high degree of convergent evolution of the osseous patella in mammals until the biomechanical function and genomic control of the patella are better understood, and improved phylogenetic sampling improves resolution of when it evolved in particular lineages.

## Patellar developmental genetics

Molecular phylogenomics provides a potential independent or synergistic approach to resolving issues of patellar evolution. If specific genomic sequence signatures could be associated with patellar status, then comparison of the genomes of the various extant but widely separated groups with a bony patella might indicate whether these represent convergence events or a common ancestral event (i.e. identified via shared evolutionarily transmitted genetic markers required for patellar development). For example, it has recently been shown that the ability to taste sweet carbohydrates in hummingbirds represents a trait convergence. Hummingbirds diverged from the insectivorous swifts, in which the sweet taste receptor is inactivated by mutations in the receptor coding gene. In hummingbirds, the ability to taste sweet has been re-acquired, apparently through molecular adaptation of the umami receptor to detect sweet molecules (*Baldwin et al., 2014*). It would be helpful to understand the (developmental) genetics of the patella as a step toward the identification of such sequence signatures. Developmental genetic studies in two mammals, humans and mice, have identified genes required for correct patellar specification. The known functions of some of these genes are informative regarding their requirements.

There are currently approximately 12 human genetic disorders with identified molecular bases that regularly include abnormal, reduced or absent patellae (hypoplasia or aplasia) as an important aspect of the phenotype (reviewed by *Bongers et al. (2005)*, see also *Warman et al. (2011)* and Table S2 for details). There are also several genes whose genetics in mice indicates relevance to patellar development at least in rodents. A detailed discussion of all these syndromes and genes is beyond the scope of this study. However, the known patella-related genes can be broadly organized according to three major developmental processes: limb specification and pattern formation (transcription factors such as *LMX1B*, *TBX4*, *PITX1* and mouse *Hoxaaccdd-11, SOX11* and signalling factor *WNT7A*); bone development, biochemistry and regulation (*GDF5*, *CHRNG*, *SLC26A2*, *COL9A2* and *AKT1*); and genes involved in DNA replication and chromatin (*ORC1*, *ORC4*, *ORC6*, *CDT1*, *CDC6*, *GMNN*, *CDC45*, *RECQL4*, *KAT6B* and *ESCO2*). Of these, the genes of replication and chromatin are the most unexpected, and potentially of the most interest for evolutionary studies. Patellar ossification may be dependent on the timing of DNA replication in particular cells, or else may be affected by aberrant gene regulation resulting from mutations in replication and chromatin factors. In either case,

the target genes mis-regulated in these syndromes, if they can be identified, may provide useful evolutionary markers to distinguish convergent from homologous patellar status.

Developmental studies in mouse or chick embryos, sometimes with induced paralysis, document the additional importance of local environmental factors in patellar ontogenesis (*Hosseini & Hogg, 1991*; *Mikic et al., 2000*; *Nowlan et al., 2010a*, *2010b*; *Osborne et al., 2002*; *Rot-Nikcevic et al., 2006*). Similarly, embryonic development and hindlimb activity in the case of particular marsupials may be important in understanding the diversity of patellar states in this group. A better understanding of these environmental processes will also be helpful to disentangle genomic versus epigenomic regulation of patellar development, and hence evolution.

# CONCLUSION

## How "the mammalian patella" evolved

The widespread, repeated evolution of the bony patella across evolution argues for an important role in locomotor biomechanics. In animals lacking an ossified patella (e.g. Lissamphibia, Testudines, Crocodylia; as well as many extinct lineages of tetrapods), the consequences of this ancestral absence for hindlimb function remain mostly unstudied. This mystery is striking, in particular, within Mammalia where most marsupials lack an ossified patella, as did numerous fossil stem-mammals, despite seeming to share common ecological niches and the associated locomotor requirements. This sporadic occurrence in marsupials and stem mammals contrasts with its near universality and evolutionary stability in the Eutheria as noted above.

The exact number of independent origins of a bony patella among mammals remains unclear, but we have estimated at least four convergent episodes inside Mammaliaformes, and several instances of patellar "loss" (with apparent re-gain in some marsupials). The pattern of acquisition and loss will require revisiting as new fossil material is discovered, as our evolutionary reconstructions are dependent on single specimens for many ancient taxa. Moreover, patellar status has not been verified for all >5,000 eutherian and >330 metatherian species (*Wilson & Reeder, 2005*), so it is possible that additional placental species (other than the fully aquatic forms) may be found lacking, or marsupials having, a bony patella. A recent evolutionary study documented many apparently independent evolutionary origins of the caecal appendix in mammals; thus the convergent evolution of unusual anatomical structures like the osseous patella has precedent (*Smith et al., 2013*). Similarly, blue colouration among tarantula spiders apparently involved at least eight independent evolutionary acquisitions, among different microscopic anatomical structures affecting spectral reflectance and hence general external colour (*Hsiung et al., 2015*). A better understanding of the genomic signatures required for development of such novel structures should be very helpful to deconstruct the observed complex patterns of evolution, distinguishing between convergent evolution (homoplasy) and shared inheritance (synapomorphy/homology).

Given that the patella evolved, and was also lost, multiple times in mammals and other Tetrapoda (Fig. 3), one thing is clear. Much as we have referred to "the patella" throughout

this study, there is no such thing—perhaps not even a single "mammalian patella". The story of patellar evolution is one of many (bony) patellae; a story of diverse evolutionary origins as well as forms, functions, ontogenies and perhaps even diverse underlying genetics. *Mottershead (1988)* wondered if the patella is "not typical of its kind" for a sesamoid bone (*Mottershead, 1988*). Yet even patellae are not necessarily typical for patellae, let alone other sesamoids—there are double or fatty patellae in some birds (*Regnault, Pitsillides & Hutchinson, 2014*), proximal suprapatellae and/or fibrocartilaginous patelloids in many marsupials, no ossified (or even other forms of) patellae in many species, and even amongst those animals that have patellae, there are numerous shapes and sizes of patellae (Fig. 4; Figs. S1–S3), suggesting still-unappreciated lifestyle constraints in patellar (and knee joint) mechanics.

While we have provisionally used the terms "patelloid" and "suprapatella" for non-ossified tissues near where the patella is or might be found, the validity of these terms needs further inspection in a broader context. Certainly, patellae exist in non-ossified forms in younger animals before endochondral ossification completes, and where such ossification does not initiate at all during ontogeny it may be best to apply the term "patella" to such tissues rather than invoke new terms for the same organ that simply underwent different tissue development; as above, a case of divergent character state transformation rather than distinct characters (i.e. new organs). This is not simply a semantic issue as the implications for evolutionary novelty, adaptation and "evo-devo" of patella-like structures will depend on the decisions made about homology of these traits in organisms, and how those decisions are communicated by the choice of anatomical terminology.

## Future prospects

Our discussion of patellar evolution in Mammalia has identified several areas where key questions remain unresolved, in addition to uncertainties about the amount of convergence/parallel evolution in origins of the osseous patella and about specific roles of (and interactions between) genetic/developmental factors in bony patellar formation/loss. Considering that mechanical loads are known to play an important role in the development of sesamoid bones (in particular in early ontogeny), studies linking these loads to genetic/developmental control as well as broad evolutionary patterns could prove very insightful, especially in explaining the seemingly large amount of patellar homoplasy in mammalian evolution. Mammals may be less sensitive (i.e. more genetically assimilated e.g. *Vickaryous & Olson, 2007*) than birds in terms of the relative influence of mechanical loads on bone (including sesamoid) ontogeny (*Nowlan et al., 2010b*)—this idea deserves better testing as insight into load-based influences improves. Furthermore, indications that some bones within an organism may be more responsive to their loading regime (*Nowlan et al., 2010a*) may be of great relevance to interpreting patellar biology and evolution, but at present strong inferences cannot be drawn about how variable the patella's responsiveness to mechanics is within or among organisms. There is clearly much room for further study of the patellae of mammals and other tetrapods, and here we have noted directions in which these might most beneficially be directed.

## ACKNOWLEDGEMENTS

Tremendous thanks are due to the many investigators who have generously given their time and energy to educating Mark E. Samuels in the fine points of hindlimb anatomy and tetrapod evolution. Special thanks to John Wible, Zhe-Xi Luo, Maureen O'Leary, Daniel Gebo, Emmanuel Gheerbrant, Jin Meng, Pascal Tassy, Cyrille Delmer, Salamet Mahboubi, Xijun Ni and Bruce Shockey for help with information on early mammals, Bonnie Smith, Norberto Giannini and Bill Schutt for help with bats, Mike Archer, Vera Weisbecker, Robin Beck, Tom Grant and Kurt Galbreath for help with monotremes, Guillermo Rougier, Jose Bonaparte, Eric Sargis, Anna Gillespie, Analia Forasiepi and Natalie Warburton for help with marsupials, Cheri Deal, David Skidmore, Judith Hall, Philippe Campeau and Les Biesecker for information on patellar genetics, Virginie Millien and Anthony Howell for access to and discussion of materials at the McGill University Redpath Museum of Natural History in Montreal, and Michel Gosselin for access to and documentation of materials at the Canadian Museum of Nature in Ottawa. We thank Matthew Lowe (curator, University Museum of Zoology, Cambridge, UK) for access to osteological specimens of mammals. We also thank numerous reviewers of earlier versions of the manuscript for many valuable suggestions.

### Funding

Mark E. Samuels was supported by the Centre de Recherche du CHU Ste-Justine. John R. Hutchinson was supported by a Senior Research Fellowship from the Royal Society and Leverhulme Trust in 2012, and grant number RPG-2013-108 from the Leverhulme Trust. Sophie Regnault was supported by a PhD studentship from the Royal Veterinary College. The funders had no role in study design, data collection and analysis, decision to publish or preparation of the manuscript.

### Grant Disclosures

The following grant information was disclosed by the authors:
Centre de Recherche du CHU Ste-Justine.
Senior Research Fellowship from the Royal Society and Leverhulme Trust in 2012, and grant number RPG-2013-108 from the Leverhulme Trust.
PhD studentship from the Royal Veterinary College.

### Competing Interests

John R. Hutchinson is an Academic Editor for PeerJ.

### Author Contributions

- Mark E. Samuels conceived and designed the experiments, performed the experiments, analysed the data, contributed reagents/materials/analysis tools, wrote the paper, prepared figures and/or tables and reviewed drafts of the paper.

- Sophie Regnault conceived and designed the experiments, performed the experiments, analysed the data, contributed reagents/materials/analysis tools, wrote the paper, prepared figures and/or tables and reviewed drafts of the paper.
- John R. Hutchinson conceived and designed the experiments, performed the experiments, contributed reagents/materials/analysis tools, wrote the paper, prepared figures and/or tables and reviewed drafts of the paper.

## Data Deposition

The raw data has been supplied as Supplemental Dataset Files.

## Supplemental Information

Supplemental information for this article can be found online at http://dx.doi.org/10.7717/peerj.3103#supplemental-information.

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
