# Peer review of "Evolution of the patellar sesamoid bone in mammals"

_PeerJ, doi:10.7717/peerj.3103_

## Round 0.1 · original submission · Major Revisions

I have now received two reviews of your paper, both reviewers concur in that your manuscript is worth publishing. I think that by following their suggestions it will be further improved.

I also think that this is a vary valuable contribution to our knowledge of the patella sesamoid. However, I would like to bring your attention some issues that needs further clarification. In this manuscript you stressed that: Sesamoids are best defined as “skeletal elements that develop within a continuous band of regular dense connective tissue (tendon or ligament) adjacent to an articulation or joint” (Vickaryous & Olson 2007); and in Regnault et al. 2016 (J. Morphol) it was stressed that “It must be noted that the term “sesamoid” may refer to any organized, intratendinous/intraligamentous structure including those composed of fibrocartilage (e.g., the cartilago transiliens; see Tsai and Holliday, 2011”. It is therefore hard for me to understand why is your definition of the patella restricted to an osseous sesamoid. Could you please explain me what is your homology criterium? In my view, if we have a skeletal structure such as a fibrocartilage or cartilage, embedded in the patellar tendon, there is no reason to does not recognize it as a patella. As this is an organized intratendinous structure, located in the quadriceps tendon, it fits the sesamoid definition, and fits the patella definition. Even though you have clearly stated that your interest relay on the osseous patella, I would be grateful of knowing your perspective about this point. Additionally, you coded the fibrocartilaginous “patelloid” as an alternative for absent patella, and I totally agree with your coding. I just would also call patella to the fibrocartilaginous sesamoid. In line with this, I disagree with the idea that only a bony patella is a true patella. I do not think that a fibrocartilaginous patella is a false one, is probably a patella responding to a different kind of stimulus.

With respect to this paragraph:

The patella is the only sesamoid bone counted regularly among the major bones of the human body (Vickaryous & Olson 2007)…
I would like to point out that, although small, the pisiform is another putative sesamoid counted regularly among the bones of the Amniota.

As to the absence of the patella in Lissamphibia, there is at least one record of this sesamoid in the paper from Hebling et al. (2014) Morphological modifications of knee articular cartilage in bullfrogs (Lithobates catesbeianus) (Anura: Ranidae) during postmetamorphic maturation. Zoomorphology, see their Fig. 3A. They do not discuss the sesamoid presence but I agree with their identification. Our group have a manuscript under review reporting the presence of a fibrocartilaginous patella in several frog species. I think that it would be very interesting if you could consider these data in the optimization of your figure 3.

minor:

pg. 23 ln. 581 delete italics in Argot 2004
pg. 23 ln. 582 insert dot after (C. Argot, pers. Comm.)
pg. 24 ln. 615 “but a fibrocartilaginous or otherwise soft tissue composition of the patella (coded as state 1)...” I think that this is what I mean. It can be another tissue composition of a patella (an also true patella). However, you need to be consistent, in your view, is it a patella or a patelloid? I am well aware that more developmental data are needed, but I am just referring to the name that you use since in ln. 622 you call again patelloid to a primarily fibrocartilaginous structure in place of an ossified patella.
ln. 886-897 I find this conclusion a little disturbing. I rather prefer to believe that there is such a thing like a patella embedded in the tendon of the extensor femoris of most tetrapods, but this is maybe because I am an old-fashioned anatomist.
ln. 892-893 ...there are double or fatty patellae in some birds (Regnault et al. 2014)… Yes, I concur, no matter what is its tissue structure, then you can identify a fatty patella.

Reviewer 1 ·

Basic reporting

Minor:

Where the authors discuss various human genetic disorders (line 830-846) they might want to consider adding relevant references for each gene. Since these genes and their relations to specific human disorders are also summarized in a supplementary table they could add the reference there and also add the OMIM id for each human condition.

Experimental design

Minor:

Overall, the analysis of the data is very intuitive and interesting. However, as a reader who is not well acquainted with phylogeny trees assembly I believe I could have benefitted from a more specific (or graphical) explanation of the pipeline used to construct the trees; alternatively, how the Mesquite software was used and the models chosen. This could also be added as a supplementary.

Validity of the findings

No comments

Additional comments

no comments

Reviewer 2 ·

Basic reporting

Photographs emphasizing the knee region in Figures 4 and S2 can benefit from labels for bones, patella, the patellar tendon, and epiphyses (if unfused). This is necessary because the hind limbs are poised at different degrees of flexion and orientation. Additionally, please specify if any of the display specimens are fossil casts rather than original specimens.

Experimental design

This study reconstructs the evolutionary transitions of the patella within the mammalian lineage using multiple character optimization techniques, and comments upon the developmental origins and possible functional significance of the patella within each lineage. The manuscript is self-contained, and the authors’ attention to details and rigorous examination of each clade is commendable. However, two issues deserve attention before the manuscript should be considered for publication:
1. The authors performed ancestral state reconstructions on Mesozoic Mammaliaformes, Metatheria, and Eutheria using both parsimony and maximum likelihood methods, with multiple alternative tree topologies used to account for contentious nodes. However, it is not clear if the three clades were included in a single tree for analyses, with separate trees in Figs. 5-7 for the sake of presentation, or rather did the author separately analyzed three distinct trees, with the ancestral state recovered from Mesozoic Mammaliaformes (Fig. 5) serving as the ancestral state for the two sub-clades respectively. Please clarify and justify
2. The author stated that parsimony is used only as an illustrative way of presenting general patterns in patellar evolution. This is appropriate for the generalized phylogeny in figure 3. However, given the substantial phylogenetic information given by branch lengths and divergence dates, it may be more appropriate to use maximum likelihood trees for figures 5-7, with relative likelihoods stated for contentious nodes.

Validity of the findings

No Comments.

Additional comments

No Comments

---

## Round 0.2 · accepted · Accept

Thank you for considering the suggestions of the reviewers and mine. This is a valuable contribution to our knowledge of sesamoids.